# A universal error source in past climate estimates derived from tree

# rings

Juhani Rinne<sup>1</sup>, Mikko Alestalo<sup>1</sup> and Jörg Franke<sup>2,3</sup>

<sup>1</sup>Finnish Meteorological Institute, P. O. Box 503, FI – 00101 Helsinki, Finland

<sup>2</sup>Institute of Geography, University of Bern, 3012 Bern, Switzerland
 <sup>3</sup>Oeschger Centre for Climate Change Research, University of Bern, 3012 Bern, Switzerland

Correspondence to: H. J. Rinne (juhani.rinne@kolumbus.fi)





Abstract. Recently it has been shown that climate estimates derived from tree rings often tend to show erroneous long-term oscillations, i.e. there are spectral biases at low frequencies. The result is independent of parameter studied (precipitation or temperature) or measured proxy (tree ring widths or maximum latewood densities). In order to find reasons for such universal errors, a new reconstruction method is introduced where no age dependence of the tree rings is determined. The

- 5 aim, however, is not to generate better reconstructions but to study error variances of long-term oscillations. It is shown that paucities and data gaps due to missing trees increase the risk for erroneous low-frequency variability. A general approximate formula is introduced in order to estimate the presence of such a risk. A case study using Torneträsk data from Northern Sweden illustrates how longer periods with missing trees cause paucities and gaps leading to erroneous climatic oscillations. Systematic underestimation of the temperature around AD 1600 and after 1950 ("divergence") is in the study case explained
- by such data gaps and paucities. 10

Key words: Erroneous oscillations in climate reconstructions - Tree rings - Tornedalen temperatures - Torneträsk temperatures

#### **1** Introduction

Tree-ring chronologies, as they are used for climate reconstructions, are typically constructed by averaging measurements 15 from multiple trees for each year (Cook and Kairiukstis, 1990). The idea behind it is that generating a mean will cancel out the random noise recorded by the individual trees and that the remaining variations represent the common signal, e.g. climate. However, merging all measurements for each year is also a necessity because every year a varying subset of single trees is available. The age structure of trees in each year is likely to change, especially further back in the past, when fewer samples are available or in the last decades, if scientists did not sample very young trees (Fig. 1). 20

The single tree measurements, for climate reconstructions mostly tree-ring width (TRW) or maximum latewood density (MXD), show an age-growth relationship, so that young trees generate wider rings than older trees. These effects are usually removed by subtraction of an average growth function. If there are climatic trends or low-frequency variations (in this study used for multi-decadal to centennial variability) it has been difficult to separate this variability from the growth

```
25
  trend and to keep the low-frequency changes in the remaining residuals.
```

The dendrosciences community has been well aware of these challenges (e.g. Cook et al., 1997). They developed methods such as regional curve standardization (RCS) to preserve low frequency variability in the age-detrending process (e.g. Mitchell, 1967; Briffa, 1996; Esper et. al., 2003). Additionally, the response of young trees to climate may differ from that of old trees. Also, the seasonality of signals or the sensitivity to temperature or precipitation limitation may change over

time (Esper et al., 2008). 30

> Despite all efforts it remains uncertain if most RCS based reconstructions show the correct amount of low frequency variability. In fact, Franke et al. (2013) succeeded in showing that many present day state-of-the-

art reconstructions still tend to be biased in the low frequency part of the spectra. To uncover potential reasons of the bias and to make case studies requires new approaches. In order to study the bias in more detail, we introduce a new method of building tree-ring chronologies in this study. This method does not rely on detrending for age-growth relationships and simultaneously allows for low-frequency error estimation.

5

20

In the following chapter, we introduce the new method theoretically. It is to point out that the aim is not produce better temperature reconstructions. Instead, erroneous long-term oscillations and their reasons are studied. An illustrative case study is based on the well-known tree-ring MXD data from Torneträsk (Schweingruber et al., 1988) because this temperature reconstruction is known to show erroneous long oscillations (Rinne et al., 2014).

#### 2 Methods

# 10 2.1. Preliminary considerations of the computations

The measurements (e.g. maximum densities of the tree rings) are given by M(b,t), where *b* and *t* are the age class of the tree ring and the calendar year (AD 441-1980), respectively. The age class refers to the cambial age of the ring of any tree. The unit of b is year following the convention. Thus, if *b*=1, the tree ring is the first one from the pith.  $b=b_{max}$  is the highest age class accepted into the computations. The measurements available during any calendar year *t* are  $M(b_b t)$ , where *i*=1,2,3,..., *n* 

and  $b_i 

5

## 2.2 Explicit computation formulae

Let  $M_i$  and  $M_{i+1}$  be measurements of two successive age classes  $b_i$  and  $b_{i+1}$  in the year to be studied. The missing values between them are interpolated and then summed up from  $b_i$  to  $b_{i+1}$ . That sum is here linearly approximated by  $(b_{i+1}-b_i)(M_i+M_{i+1})/2$  (following the trapezoidal rule, Fig. 2). The sum of all such intermediate sums divided by  $b_n$ - $b_1$  will give the average of all measurements, interpolated or observed, between  $b_1$  and  $b_n$  as follows

$$M_{int} = \sum_{i=1}^{n-1} \frac{(b_{i+1} - b_i)(M_i + M_{i+1})/2}{b_n - b_1}$$
(1)

Assume for the time being that the tree ring measurements would decrease linearly with the age so that the observations of the youngest and oldest age classes of  $b_x$  and  $b_y$  are  $M_x$  and  $M_y=M_x-k(b_y-b_x)$ , respectively, where k is a constant. Then the mean value of those observations is  $(M_x+M_y)/2$  or  $[M_x+M_x-k(b_y-b_x)]/2$ . Suppose now that measurements only until the age class  $b_z < b_y$  are known. Then the oldest known measurement is  $M_x-k(b_z-b_x)$  and the average of all known measurements becomes  $[M_x+M_x-k(b_z-b_x)]/2$ . Thus by deleting older observations  $b_z+1$  through  $b_y$ , the estimate of the average will be increased by  $k(b_y-b_x)-k(b_z-b_x)=k(b_y-b_z)$ . Experimenting in this way with the real data it was found that such

a deletion indeed tends to change the true average almost linearly. Accordingly, to take into account the effect of the missing youngest and oldest age classes and to extrapolate their contribution, the average of the known measurements must be corrected by adding Myoung≈r1(b1-1) and Mold≈rn(bn-bmax), respectively, where r1≈0.000128 and rn≈0.000170 are determined experimentally. Note that Mold results in a negative correction. The accuracy of such corrections turned out to be low their variances being roughly [r1(b1-1)]<sup>2</sup> and [rn(bn-bmax)]<sup>2</sup>. The computations are illustrated for an individual year in Fig. 2 for bmax=300. The extrapolation corrections are more complicated if most of the older age classes are unknown (bn<30) or the impact of the missing youngest age classes (b1<≈20) is to be estimated. However, to avoid too complicated formulae, in these exceptional cases the simpler formulae given will here be applied. By adding the corrections, the yearly</li>

25 
$$M_{ave} = M_{young} + M_{int} + M_{old}$$
(2)

average of the interpolated, extrapolated and measured values becomes approximately

 $M_{int}$  in fact presents a weighted average of the measurements available. It can be seen by reordering the terms of  $M_{int}$  as follows

$$M_{int} = \sum_{i=1}^{n} \frac{(b_{i+1} - b_{i-1})M_i / 2}{b_n - b_1}$$
(3)

where use of two formal additional parameters is made, namely  $b_0=b_1$  and  $b_{n+1}=b_n$ .

#### 2.3 Error estimation

5

25

errors.

Assuming that the errors in the measurements in Eq. (3) (being from different trees) are independent of each other and that their error variance is  $S^2$ , the error variance of the sum  $M_{int}$  becomes from Eq. (3) as follows

$$Var_{int} = \sum_{i=1}^{n} \frac{S^2 (b_{i+1} - b_{i-1})^2 / 4}{(b_n - b_1)^2}$$
(4)

#### 10 Adding error variances arising from the extrapolations, the error variance of the final average becomes

$$Var_{ave} = r_1^2 (b_1 - 1)^2 + \sum_{i=1}^n \frac{S^2 (b_{i+1} - b_{i-1})^2 / 4}{(b_n - b_1)^2} + r_n^2 (b_n - b_{\max})^2$$
(5)

The variance estimate consists now of two kind of factors, those unrelated  $[r_1^2, S^2, r_n^2]$  and those related  $[(b_1-1)^2, \sum (b_{i+1}-b_{i-1})^2 (2(b_n-b_1)]^2, (b_n-b_{max})^2]$  to the distribution of age classes. To equalize their dimensions, they are multiplied and divided by  $b_{max}$ , respectively and can be rewritten as to  $[(b_{max} r_1)^2, S^2, (b_{max} r_n)^2]$  and  $\{[(b_1-1)/b_{max}]^2, \sum (b_{i+1}-b_{i-1})^2/[2(b_n-b_1)]^2, [(b_n-b_{max})/b_{max}]^2\}$ . The latter group is now dimensionless and describes relative variance terms. They present a new kind of error source, which depends on the age classes, i.e. only on the distribution of the measurements over the trees and years (Fig. 1).

It has been assumed that the yearly measurements do not overlap  $(b_i 

To study the impacts of the paucities and data gaps is tedious on the basis of Eq. (5) or graphs like in Fig. 3d. The coefficients  $r_1$ , S and  $r_n$  are unknown and the corresponding terms must be studied separately. In the following they are combined by making an *ad hoc* assumption that  $r_1=r_n=0.005$ S. Then the variance estimate can be written as follows

5 
$$Var_{ave} \approx S^2 Var_{rel}$$
 (6)

where the relative, dimensionless variance term depends only on the distribution of the age classes over the trees and years as follows

$$Var_{ave} = 0.7^{2} \{ 0.005^{2} (b_{1} - 1)^{2} + \sum_{i=1}^{n} \frac{(b_{i+1} - b_{i-1})^{2} / 2}{(b_{n} - b_{1})^{2}} + 0.005^{2} (b_{n} - b_{max})^{2} \}$$
(7)

|   | ~  |  |
|---|----|--|
| L | 11 |  |
| r | v  |  |
|   |    |  |

15

The additional scaling factor  $0.7^2$  is explained in the following.

If S<sup>2</sup> represents the variance of random errors of the observations and their distribution is a Gaussian one, the variance of the average of the observations can be estimated from  $S^2/N$ , where *N* is the number of observations. In Eq. (6) it is represented by  $1/Var_{rel}$ . Accordingly,  $1/Var_{rel}$  can thus be seen to represent the degrees of freedom (DOF). These are shown in Fig. 3c. The scaling factor (here  $0.7^2$ ) is to be selected so that DOF are as high as possible but not exceeding *N*. Note, it is only necessary to know the actual distribution of the age classes (Fig. 1) to derive the DOF by approximation. The DOF can be essentially seen as a parameter that controls the effect of the paucity and data gaps of the original data on

the final reconstruction.

The degrees of freedom,  $1/Var_{rel}$ , are designed to work in data periods with data gaps and paucities. Their use is especially recommended as a warning tool in such cases of missing data.

#### 3 Torneträsk case study

The classical record of MXD [g/cm<sup>3</sup>] from Torneträsk consist of 65 trees grown between 441 to 1980 AD (Schweingruber et al., 1988). The scaling from MXD to temperatures can be achieved by multiplying the MXD 25 measurements by 15.92 (Rinne et al., 2014). The same scaling factor will be used here. Thus, no explicit calibration against the instrumental observations was necessary. The temperature reconstruction (hereafter b270 refers to the selection of  $b_{max}$ =270) is shown in Fig. 3a.

To further illustrate the performance of our reconstruction method, a comparison with instrumental observations for the summer months (JJA) roughly after 1800 will be made. As the focus is in the erroneous long-term biases, smoothing will

30 be applied. The case is challenging because of a wide data gap due to the missing trees during recent years (Fig. 1).



Klingberj and Moberg (2003) composited a series of instrumental observations made in the Tornedalen region, some 300 km from Torneträsk. Their construction begins with instrumental data from Övertorneå (1802-1838). The observation hours were not stated explicitly for 1826-38 and the authors had to assume them. Here we correct that assumption by increasing their JJA temperature reconstructions by 1°C. The procedure can be interpreted as if an unknown component in the instrumental observations had been deduced from the tree ring estimates. Nevertheless, temperatures

before ca. 1850 may still contain biases (Melvin et al., 2013; Grudd, 2008).

Our method allows to choose  $b_{max}$  optimally for each consideration. For the full period  $b_{max} = 270$  was seen to be the optimal selection. However, for the shorter, more recent period 1802-1980  $b_{max} = 370$  seems to be better. Such a choice makes it possible to include into the computations numerous older tree rings available during recent decades (Fig. 1). As a result, the systematic underestimation becomes lower (Fig. 3a).

The resulting tree-ring reconstruction is shown in Fig. 4 by adjusting its mean to the observation mean during 1850-1950. Otherwise no calibration against observations is made. Instead, the scaling factor of 15.92, selected beforehand, is applied. The weather variations are outside of the scope of the present work so that both curves have been smoothed (26years cubic splines). Note here that the geographic areas of the estimates and observations differ somewhat. Taking into

account the high number of missing younger age classes, the temperature estimate during the recent years can be seen to be satisfactory, too.

#### 4 Discussion

#### 4.1 Bias of the reconstructions

In the tree-ring studies the age-growth function is in a central role. It is a problematic case if measurements deviate significantly from the age-growth function applied, i.e. there is a tree with biased growth. In our approach the age-growth function is by-passed. For comparison, in Fig. 3a the Torneträsk reconstruction made in Briffa et al. (1992) using the classical approach is shown together with the present approach using the same Torneträsk data.

The reconstructions in Fig. 3a are astonishingly close to each other, in spite of the very different computational methods. All cases show a (biased) quasi cycle of a wave of ca. 350 years in length. Importantly, the bias is not in the tree

ring measurements but in the temperature estimates and is due to missing trees. Melvin et al. (2013, their Fig. 4) present several variants of the reconstructions, including those derived from the tree ring widths. All of them show a similar biased structure as long as only the Torneträsk data is included. This is because they are all derived from the data with the same structure (Fig. 1).

#### 30 4.2. Explanation of the bias observed

following discussed in detail.

5

20

The yearly variations of the dimensionless variance terms for the Torneträsk case are shown in Fig. 3d. Note that they are relative error terms and the final error variance depends on the dimensional portions  $(b_{max} r_l)^2$ ,  $(b_{max} r_n)^2$  and  $S^2$ . These include in a compressed form the error variances due to random variations in the measurements. Paucities in the tree population or periods of missing trees (Fig. 1) cause successively data gaps in young, intermediate and old age classes leading to variance variations in Fig. 3d. The increased variance decrease the degrees of freedom and so makes the bias more probable, which is then seen as erroneous long-term oscillations in the reconstructions. This chain of events is in the

To further study the connection between the relative error variances and long-term error, the bias is estimated with the aid the temperature estimate in Esper et al. (2012). The original Torneträsk data used in this study consist of 65 trees.

- 10 The yearly numbers of tree-ring measurements may vary strongly being sometimes very low (Fig. 3c) and never over 25. However, when the degrees of freedom in Fig.3c are e.g. over 14, the bias nearly vanishes. In Esper et al. (2012), the number of trees is as high as 578. Between 650 and 2000 AD the numbers of yearly measurements are only exceptionally less than 25, normally clearly more (their Fig. 2). Accordingly, it can be expected that the degrees of freedom are high enough so that their reconstruction can be seen to be more bias free. A more detailed evaluation could be done following our
- Eq. 7 but it is beyond the scope of this study. Thus the difference between the b270 reconstruction and the one in Esper et al.(2012) can be seen to represent approximately the temperature bias (Fig. 3b) due to the Torneträsk data.

Fig. 3b shows how the long-term variations in the reconstructions (Fig. 3a) are mostly erroneous. An explanation of this bias, derived from the presented theory (Eq. 7), is given in Fig. 3c. Some minima and maxima of the DOF are indicated by full and open circles, respectively. It is seen, that strong minima (maxima) of the DOF well predict following biased (unbiased) values which are reflected in reconstructions (Fig. 3a).

To illustrate, a population change is seen in 1356 AD (Fig. 1) where the oldest age class (<270) drops suddenly down to b=116. As a consequence, variances of the older and intermediate age classes are peaked in Fig. 3d, further lowering strongly the DOF in Fig. 3c. This is reflected in the bias (Fig. 3b) and in the reconstruction (Fig. 3a, smoothed values in Figs. 3a and 3b naturally lag the sudden changes in Figs. 3c and 3d).

25 Correspondingly the optimal distribution of the measurements around 869 and 1500 AD (Fig. 1) is reflected in low error variances (Fig. 3d), high number of the DOF (Fig. 3c) and unbiased temperature estimates (Fig. 3b).

Low values of the DOF (<5) are related to extrema of the bias. High values of the DOF (>14) indicate vanishing bias. The latter ones are seldom and therefore bias-free cases are seen only temporarily in Fig. 3b.

In Fig. 1, there is a repetitive similarity between the two longer periods with nearly continuous samples of trees (AD 1250-1500 and 1600-1800) and two periods without new trees (AD 1500-1600 and 1800-1980). The corresponding terms of the relative variance (Eq. 5) will be discussed in the following.

The large first peaks of variance around 1356 AD (Fig. 3d) result from missing intermediate (interpolated) and old age classes. These damp slowly out and new peaks are seen at 1600 AD. Here they are due to the intermediate (interpolated) and young age classes. The same structure is repeated between 1600-1800 AD and 1800-1980 AD. In this way

the paucities due to missing trees in the data cause alternating data gaps leading to varying behavior in temperature reconstructions. Especially biases around 1600 and 1950 can be explained by the error terms connected to the missing younger age classes (due to the missing trees). The latter case is known as "divergence", a systematic underestimation of the temperatures (Briffa et al. 1998, D'Arrigo et al. 2008). As there is no formal difference between the cases of 1600 and 1950,

it is natural to refer to "divergence" in both cases.

In summary, the long-term biased oscillations originate from the unsuitable distribution of the measurements over age classes and years and are independent of the reconstruction methods and the existing measurements

#### **5** Conclusions

The presented method to estimate past temperatures from tree ring measurements is a new approach, where no age dependence of the tree rings is estimated. The method allows to estimate systematic errors due to the uneven distribution of the measurements caused by missing trees, i.e., by data gaps and paucities in the data. Because the structure of the tree populations varies slowly, such errors are mainly seen as long oscillations.

In the Torneträsk case studied, the biased temperature oscillations were well connected to the theoretical error terms derived. Especially the "divergence" cases (a systematic underestimation of the temperatures) around 1600 and 1950 could be explained by the error terms connected to the paucities in the data seen as missing younger age classes.

Several Torneträsk studies presented in the literature, in spite of clear differences between the different approaches, suffer similarly of paucities in the data. All of them show similar spurious low-frequency oscillations.

Franke et al. (2013) observed that the low-frequency biases can be present in studies applying maximum densities of the tree rings as well as those studying tree ring widths. The bias can further be present both in estimates of precipitation

and temperature. The presence of the low-frequency error was independent of the methods applied or aim of the work. This study strongly suggests that the paucities in the data are one explanation for such a general presence of the error. The approximate method derived here to estimate the degrees of freedom can be used to trace the potential impact of such paucities.

#### Data availability

The Torneträsk tree ring data (Schweingruber et al., 1988) applied here is available at https://www.ncdc.noaa.gov/paleo/study/4684. The Torneträsk temperature reconstruction in Briffa et al. (1992) is taken from Melvin et al. (2013). The temperature reconstruction made in Esper et al. (2012) for Northern Fennoscandia is from http://www.geo.uni-mainz.de/1358.php.

## Acknowledgements

The writers wish to thank Dr. Markku Kangas for his advice in editing the text.

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

# 10