# Peer review of "A universal error source in past climate estimates derived from tree"

_Climate of the Past, 2016_

## Short Comment (SC1) · 24 Mar 2016

The title should definitely be changed because it is really misleading. We get the impression that all tree-ring-based reconstructions are erroneous (see the words "A universal error source"). Instead they conclude that "However, when the degrees of freedom in Fig.3c are e.g. over 14, the bias nearly vanishes". Moreover they use a tree-ring-based reconstruction with larger replication (Esper et al. 2012) to define their bias.

---

## Author Comment (AC1) · 1 Apr 2016

Thank you for commenting a central point in our paper.

The error source is present in every analysis at least in the beginning of the data period. In our case that can be seen in Fig. 3c, where the degrees of freedom are low during 500-800 AD though the number of measurements increase. When applying the reconstruction of Esper et al. (2012) in our work, the leading years (centuries) were excluded.

Following your comment, the title will, however, be changed as follows. "A widespread error source in climate estimates derived from tree rings"

References Esper, J., Büntgen, U., Timonen, M., and Frank, D.C.: Variability and ex-

tremes of Northern Scandinavian summer temperatures over the past millennia, Glob Planet Change, 88-89, 1-9, 2012.

---

## Referee Comment (RC1) · Anonymous Referee #1 · 20 Apr 2016

The manuscript touches upon an important topic within tree-ring research and climate reconstruction, namely sample replication and the ontogenetic or age trend present within tree-ring series (hence referred to as detrending methods). Especially, removing age-related trends while maintaining low-frequency climatic signals is of great interest to better understand earths climatic system. Both the fact that a new detrending methodology is presented to address this issue and a case study is provided on a widely used tree-ring record gives this manuscript great potential. However, the poor link to other detrending methods and work that has been done on other relevant tree-ring biases, weakens the message given by the manuscript although very strong statements are made. Additionally, for this method to be useful to the community more emphasis should be put on clearly explaining the steps needed to execute this detrending technique. Because of this I would like to present a few points of discussion

in relation to; a) inclusion of other detrending techniques, b) the impact of tree-ring related biases and c) expanding the methodology.

Due to both the title and the context of the paper it appears that the issues in relation to RCS deterending are present within most chronologies and has not been properly addressed. However, recently a lot of work has been done on addressing and resolving detrending related issues, while these methods and sources are not being addressed or mentioned within this manuscript. This for instance includes work on the RCS related issues in Briffa & Melvin (2011 in Hughes et al. Dendroclimatology, Developments in Paleoenvironmental Research) or the work done on comparing multiple dentrending methods and their potential to maintain low-frequency signals (Peters et al. 2015 Global Change Biology). Additionally other detrending methods like the signal-free detrending should be considered as they show great potential in being less affected by age-related issues (see Melvin & Briffa 2008 Dendrochornologia). It was also surprising to see the comparison between the proposed method and the reconstruction performed by Briffa 1992, while multiple comments and revisions have been made on this chronology (see: Melvin et al. 2013 Holocene; Matskovsky et al. 2014 Climate of the Past). The current comparison is therefore very difficult to interpret as it is not clear whether one can state that one is better than the other. Additionally, the fact that the difference between the methods is very small when the sample size is higher than 14 raises more debate on sample replication than on the relevance of the method. To validate the value of the newly proposed method it is therefore crucial to include more dentrending methods and more recent Torneträsk reconstructions.

Many biases and problem are present within tree-ring data. One of these includes the ontogenetic or age trend. However, many other biases are present like the persistent growth patterns or management related issues. Multiple of these biases have been addressed in literature and have been shown to affect low-frequency patterns, which is the main interest of this manuscript. However little attention is provided on either how these biases affect the proposed method or discussed how these could affect the low-frequency signals. Work on both sampling strategy (Nerhbass-Ahles et al. 2014 Global Change Biology) and multiple biases described in literature should be addressed as these are common problems in constructing chronologies (See: Brienen et al. 2012 Global Biogeochemical Cycles; Bowman et al. 2013 Trends in Plant Science; Groenendijk et al. 2015 Global Change Biology; Autin et al. 2015 Dendrochronologia).

For reproducibility it is essential when introducing a new method that all steps and components within the procedure are clear. In general it is therefore important that enough attention is given to the methodology section, which currently is not sufficient to apply this method on other datasets. As an example more information should be provided on how the sensitivity analysis is performed to determine the upper limit of the age classes (bmax). Additionally, how one should determine the other required parameters to perform the computation is unclear (e.g. r1 and rn). What is also vague within this method is how the extrapolation affects the results (see Figure 2). If there is for instance a year with only a few young age classes a large proportion of the mean is determined by the extrapolated older classes, which are heavily dependent upon the assumption you make within this extrapolation procedure (which, if I understand correctly, in the manuscript is proposed as a negative linear relationship which can be highly debated). How these situations affect the method should be described or analysed. Finally, the error estimation and the representation of missing age classes in Figure 3 could be of great value to the community. Especially, visually showing where specific age-classes are lacking could help to detect specific biases and to disentangle whether low-frequency signals are caused by sample replication or climate. Because of this relevance I feel more attention should be given to the methods section on this as currently the description is highly condensed and in some parts not clear.

---

## Short Comment (SC2) · 26 Apr 2016

This paper introduces a "new" standardisation method to construct chronologies. It tries to show that there is low-frequency bias in tree ring chronology reconstructions. The new reconstruction method is used with the Tornetrask MXD data from 1988 to demonstrate this bias.

The authors state that "The presented method to estimate past temperatures from tree ring measurements is a new approach, where no age dependence of the tree rings is estimated." yet they are clearly removing the age related growth of trees as a linear trend. For each calendar year they have a few measured rings (e.g. 20 values). They effectively create further values by extrapolation including younger rings and older rings to a total of 270 (e.g. adding 250 values) with a linear age-related decay. These 270

values are then averaged together thus removing the effect of ring-width aging using the presumption of some form of liner decay of ring width with age. The method of estimating R1 and Rn (page 4 line 17) will need a detailed explanation. Overall it is likely to have a similar result to that of creating and fitting a linearly decaying RCS curve. In RCS the averaging and smoothing of the RCS curve tends to reduce the climate noise from the estimation of the ageing trend whereas in the proposed method using rings from a single year which all have the same climate signal achieves this. Their conclusion that they do not remove the effect of age-related growth from their measurements is not justified as they do try to remove the age effect.

The 1988 Tornetrask MXD data were selected by age (oldest well behaved trees) from the much larger TRW data set with a view to using curve-fitting standardisation methods with sufficient replication for reconstructing medium frequency variability. An even distribution of tree rings by age in each year was not thought necessary in 1988. This is not a suitable data set to introduce or evaluate the proposed new standardisation technique.

The authors need to note that Briffa et al (2009, Hughes book chapter) show that for these MXD measurements from Tornetrask the assumption of linear decay creates a bias in the reconstruction. For TRW, the assumption of linear decay would create even more bias.

The Esper 2012 chronology has more trees (even after using mean tree rather than multiple cores) and less error due to sample count i.e. less noise. Is the age distribution of the Esper trees biased over time? No assessment is made of this so the presence or absence of systematic bias is not known and the comparison (and conclusions based on it) in this paper is not justified.

There is no attempt to distinguish between bias due to the age-related growth decay in tree measurements and noise created by poor replication and the authors confuse these two effects making their conclusions less valid and unhelpful. The presentation

in this paper is not suitable to introduce a "new" standardisation method. A comparison of new against existing methods is needed which shoulfd include a careful assessment of errors – with separation of noise related to insufficient samples and systematic bias related to poor removal of age-dependent growth and an evaluation of error magnitude. A sample data set with sufficient samples in each year to sub-divide the data and show the effect of reducing sample counts is needed and only then can the bias due to age-trend be shown.

My overall assessment is that this paper requires considerable improvement before it is suitable for publication.

Dr Thomas Melvin

---

## Referee Comment (RC2) · 27 Apr 2016

**A universal error source in past climate estimates derived from tree rings**

Juhani Rinne[1], Mikko Alestalo[1] and Jörg Franke[2,3]

[1]Finnish Meteorological Institute, P. O. Box 503, FI – 00101 Helsinki, Finland
[2]Institute of Geography, University of Bern, 3012 5 Bern, Switzerland
[3]Oeschger Centre for Climate Change Research, University of Bern, 3012
  Bern, Switzerland

Correspondence to: H. J. Rinne (juhani.rinne@kolumbus.fi)

**Article Review:** pjk-Stockholm, SE

**Introduction:**

In this study (hereafter: RN()) the authors start out to prove that paucities and age-class gaps in the Torneträsk MXD chronology (Schweingruber 1988, Briffa et al., 1992) are responsible for a low frequency (red) bias. To prove this RN() have developed a method of MXD chronology construction that purportedly accounts for paucities and age-class gaps. They evidence their method's advantage by comparing a reconstruction of the historical Tornedalen temperature record (Klingbjer and Moberg, 2003) to a reconstruction produced by their method. As evidence for the low frequency bias in the RCS chronology produced by Briffa et al., 1992 (hereafter: BF92), the authors rely on a simple visual comparison.

It is clear that much of the motivation for this study relies on the theories proposed in Franke et al., 2013; hereafter: FK2013. One cannot fail to get this message from the rather brazen title and opening sentence of the Abstract (which requires references), and the last sentence (for which the study provides no evidence). Otherwise the implied goals are commendable. It would certainly be interesting to read about a new method of tree-ring climate reconstruction that does not involve transforming raw measurements into dimensionless indices, retaining the original units (e.g., Helama, 2015), and it would certainly be informative to learn what the "true color" of an MXD chronology is. However, I am not sure the method put forth in this study does any of this; in fact I am not exactly sure what this method does other than rescales averaged, variance adjusted MXD measurements. How the results presented here lead to the conclusion there is a spectral bias, *vis-à-vis* FK2013, in the 1992 Torneträsk MXD chronology (BF92), thereby corroborating FK2013, is beyond my ability to detect. There are no spectral analyses performed, no modeling of persistence, and above all no hypothesis testing with statistical rigor of any kind. There are only graphical comparisons (wiggle matches) between chronologies and reconstructions.

In this review I will argue the paper has no merit because *i*) the study is biased and provides no proof of significance, ii) the study is based on outdated data and does not contribute to advancing knowledge, *iii*) the proposed reconstruction method is does not produce a significantly different chronology, *iv*) the method does not account for inherent growth trends in MXD data, and *v*) the validation exercises are inconclusive.

I will conclude with a summary describing what I feel is the salvageable merit of this study, and a final comment on the gap-filling procedure described within.

**Comments re: RN()**

*i*) The scientific method and researcher bias

In science the null hypothesis defines a condition or relationship that an experimenter wishes to study and test. In a quest to find a difference between two conditions, A and B, the null hypothesis would be; there is NO difference. If the conditions in question have quantities that can be measured then statistical tests are used to decide whether to accept or reject the null hypothesis. The decision to reject the null hypothesis, ergo there IS a difference, is based on the probability (significance) that the observed difference cannot be explained by chance alone. In RN() the implied null hypothesis is: the BF92 MXD chronology IS different than that produced here; however, there is no evidence that the observed difference is significant.

There are libraries full of literature describing methods of signal processing and analysis of time series (e.g., Blackman and Tukey 1958, Percival and Walden 1993, Park 1992, Thomson 1982) that one can use to describe and compare the spectral properties of two time series (e.g., FK2013). Consider, in BF92 there is a plot of the power spectra of the MXD reconstruction produced using RCS (figure 9: BF92) (fig.1). Why couldn't the current authors have done the same? Or even push the concept further and computed the cross-spectral coherence between the two chronologies (http://www.spectraworks.com/web/welcome.html).

[Figure]

Figure 1 (BF92: fig.9b: reprinted without permission). The power spectra of the MXD reconstruction "based on the 300–lag autocovariance function and individual estimates have been smoothed with the Hamming window and have 12 degrees of freedom. The null continuum and 95% significance levels (for pre–defined peaks) are also shown".

Considering the ramifications of this study, and the overt claims of "erroneous bias", one would expect to find some statistical evidence for rejecting the unbiased, null hypothesis i.e., BF92 = RN(), but we don't. This is disconcerting to me for it means there really is no hypothesis testing and it is only the investigator's word that we must accept. Given the title and the exhausting use of *bias* and *erroneous* ("bias" is used 28 times; erroneous 10), it does not take a great deal of imagination to guess what the authors will conclude. The lack of a null hypothesis, and any attempt in applying statistical rigor to results, negates the significance of the conclusions.

*ii*) Why Briffa et al., 1992?

Since 1988 when the first Torneträsk MXD chronology was developed (Schweingruber et al., 1988) there has been tremendous effort and study invested in producing millennial length chronologies and reconstructions from the Scots pine trees in Fennoscandia, particularly those surrounding Lake Torneträsk, Sweden (Esper et al., 2014 and references therein). The most relevant of these is the most recent Melvin et al., 2013 (hereafter: MK2013).

With the exception of the last ~250 years when the Schweingruber et al., 1988 measurements are updated by the addition of predominantly faster growing, young living trees (Grudd 2002, 2008), the MXD data used in MK2013 are the same as those used in BF92, and this study. In other words, ignoring the post ~1650 CE period, MK2013 is essentially a re–analysis of BF92. That being the case then the real objective experiment would be to compare the chronology and reconstruction produced by RN() to that produced by MK2013. Why this was not done is again evidence to me that there is an a priori bias in RN(). So let us do it; let's compare MK2013 with RN() and decide which has more spectral bias.

Consider figure 3 in MK2013, reproduced here as figure 2. In panel b we see the two chronologies produced from the high MXD (red) and low MXD (blue) value trees, along with their average. In addition the authors have kindly provided us with information on where in the two chronologies the sample size falls below 4 (thick and thin line widths); the source of those egregious "*biases*" the present study attempts to correct. For all practical purposes the black curve in panel c (One RCS Chronology) is effectively the BF92 chronology, and the red curve is the new, improved MK2013 Torneträsk chronology.

[Figure]

Figure 2. MK2013 main text figure 3 (reprinted without permission). "(a). The black curve is based on all samples and the curves in red and blue were built from samples with the highest and lowest values of MXD respectively, where sorting was based on comparison of mean signal-free MXD against that of a single RCS curve over their common period. b) shows mean chronologies created using two RCS curves; for high-MXD samples (red), low-MXD samples (blue), and the average of all samples (black). c) shows the chronologies created using a single RCS curve (black) and two RCS curves (red). Chronologies were low-pass filtered using a 100-year cubic spline. The thicker parts of the lines show sections of chronologies based on 4 or more samples and grey shading shows the sample counts over time." Melvin et al., 2013.

By simply comparing the red and black curves in Fig.2b one sees the new method proposed by MK2013 pays attention to temporal changes in sample depth particularly during the Medieval period where only the original Schweingruber et al., 1988 data are contributing to chronology. By visually comparing the black and red curves in Fig.2c one can imagine there is slightly less low frequency variation in red curve than the black.

*iii*) Where's the bias

"The reconstructions in Fig. 3a are astonishingly close to each other, in spite of the very different computational methods", RN(). So, is there a problem to fix? Let's consider the results of a comparison between BF92 chronology and the chronology produced by the method proposed here in RN() figure 3 reproduced below also as fig.3.

[Figure]

Figure 3. RN() Figure 3 with caption.

"Figure 3. The theoretical explanation of biased long-term oscillations between 500 and 1975 AD. Panel (**a**): Smoothed (85-year spline) temperature reconstructions derived from Torneträsk MXD data 441-1980. Shown are the estimate from Briffa et al. 1992 and the present ones both with $b_{max}$=270 and $b_{max}$=370. Panel (**b**): The smoothed (85-year spline) difference between the present temperature estimate ($b_{max}$=270) and that in Esper et al. 2012. Panel (**c**): Comparison of the sample count (number of observations) and degrees of freedom. Indicated are the seven and two cases of DOF<5 and DOF>14, respectively. Panel (**d**): components of relative variances. Some years mentioned in the text and shown in Fig. 1 are indicated in panels (**b**), (**c**) and (**d**)." RN().

In figure 3a I have marked the maximum and minimum levels of 14 major peaks and troughs in the two chronologies. For the purpose of example I used the b<=270 chronology to represent the "best" RN() chronology. It does not take a well–trained eye to see that the quasi–centennial scale variability in the b<=270 chronology is actually greater than in the BF92. For instance, between peaks 1 & 2 (fig.3a) the amplitude of the change in the b<=270 chronology is larger than in BF92, and between peaks 9 & 10, is

larger still. Therefore, if there is a spectral bias in either of these two chronologies one could easily argue that it is greater in the b<=270 series.

*iv*) The fly in the ointment

MXD measurements have inherent growth trends and are commonly detrended by computing residuals, as opposed to ratios for tree ring–width (TRW) measurements, between the measured density and some mean biological growth function, viz. eq. 1.

$$I_t = R_t - G_t \qquad \qquad eq.1$$

where R(t) is the MXD measurement for a given year, G(t) the value of some function chosen to best model the overall trend in R, and I(t) the resulting index for year t for t=1,age. As described in BF92, G(t) is commonly a smoothed version of the mean age–aligned biological growth function. In general density measurements range from .5 to .7 density units, and in their raw form MXD measurements are much more homoscedastic then TRW measurements (Figure 4). As Figure 4 clearly shows, there is still an age–related trend in MXD measurements that is not climate related and must be accounted for. [Incidentally, please check all references. I am sure the placement of Cook and Peters 1997 is not correct; as it pertains to solely TRW data and the calculation of tree–ring indices as ratios, making it tangentially relevant to our discussion here but not where it is in the main text. I believe the more appropriate reference for the main text is Cook et al.,1995)].

The fact that MXD data have demonstrably less trend than TRW measurements is the reason RN() could simply average MXD measurements and produce a plausible chronology. I suspect that had RN() accounted for the non–climatic trend in the MXD measurements then their resulting chronology would have less "bias" and quite possibly look more like BF92.

[Figure]

Figure 4. Growth trends in TRW and MXD measurements. Panel a, the age aligned trends: Regional Curves, panel b the calendar aligned trends (reprinted without permission from https://www.researchgate.net/figure/ 277853533_fig2_Figure-2-TRW-and-MXD-age-trends-a-Arithmetic-means-of-the-age-aligned-TRW-and-MXD)

*v*) Let's be fair

A perceived erroneous bias in the first multi-centennial MXD reconstruction based on RCS done 25 years ago does not demonstrate the existence of "universal error" in all RCS reconstructions. Such a statement is utterly fanciful and defamatory without proof.

The claimed illustration of bias in the Torneträsk reconstruction through comparison with the NSCAN MXD-RCS temperature reconstruction (Esper et al., 2012) in figure 3b also has problems that need to be at least admitted too. The assumption made is that the reconstruction of Esper et al. (2012) is in some sense more representative because it is based on a much larger sample size that is likely to be less affected by age-class gaps and paucities. The problem with this comparison is that the MXD data used in the Esper et al. (2012) reconstruction contains a significant amount of material from Finland, including sub-fossil lake material which the 1988 and 1992 Torneträsk chronologies do not. That is clearly indicated in Fig. 1 and Table S1 of Esper et al. (2012) and also in the main text of Esper et al. (2014).

In figure 3b, the use of completely different data, without a defendable argument for why this is okay, is not good science and undoubtedly contributes to some of the lack of agreement. In figure 3b a slightly better choice would be the N-Eur reconstruction Esper et al. (2014) as this reconstruction recognizes the affect of using trees of varying age classes.

**Summary**

Throughout this manuscript there were enough confusing comments and descriptions regarding previous work and dendrochronological methods that lead me go back and look at the author's affiliations. That's when it struck me as to what Rinne et al. () are trying to do! They are treating the Torneträsk MXD data as if the data were a group of meteorological records with periods of missing data, filling those gaps, and using the Tornedalen historical record to rescale the result. This sounds a lot to me like homogenizing data.

However, what strikes me as the most egregious non-scientific element in this work is found in following extract.

*"Klingberj* [sic] *and Moberg (2003) composited a series of instrumental observations made in the Tornedalen region, some 300 km from Torneträsk. Their construction begins with instrumental data from Övertorneå (1802–1838). The observation hours were not stated explicitly for 1826–38 and the authors had to assume them. Here we correct that assumption by increasing their JJA temperature reconstructions by 1°C.* *The procedure can be interpreted as if an unknown component in the instrumental observations had been deduced from the tree ring estimates.* *Nevertheless, temperatures before ca. 1850 may still contain biases (Melvin et al., 2013; Grudd, 2008)."*

If I understand this correctly, RN() are correcting an assumption using tree-ring estimates while at the same time claiming the tree-ring estimates are biased? This boggles my mind. Not only does this confirm my opinion on how well long-historical records can be used for the assessment of spectral bias (e.g., FK2013, Osborn and Briffa 2003), but nails the coffin shut on my opinion of the present work. It completely explains how RN() could again produce a plausible reconstruction using their method! There are so many vagaries is this story that I feel the need for a new word for bias.

I strongly disagree this work brings any new insights into the causes of spectral bias in climate reconstructions from tree rings. In fact I would argue it demeans the FK2013 argument. The appearance of long-memory in long tree-ring reconstructions is just as likely to reflect the fact that climate has varied on a variety of time scales over the past millennia (NRC 1991, Koutsoyiannis 2002) and that our extant historical records of climate are too short to model this condition. Accepting the fact that we are not likely to find any new historical records, we must do the best we can to explore the proxy record. This study does not do that at all.

The writing style and structure of the manuscript could certainly be improved. I am sure I have misunderstood a meaning or two here and there simply because I could not understand clearly what was written. The one potential merit is the interpolation/extrapolation method described in section 2.1 and illustrated in Figure 2:RN(). The question of how a non-stationary distribution of age-classes affects a chronology is one Dendrochonologists continually revisit (google any of the more recent RCS publications). I highly recommend starting with Esper et al., 2003 " *Test of the RCS method for preserving low-frequency variability in long tree-ring chronologies*". As the title implies, Esper et al., 2003 is obviously germane

to the topic in hand and should have been referenced.

The questions I ask are, what do the simulated values look like, and is this step really necessary? I wonder how biologically relevant are the simulated values? This is certainly an area in dendrochronology that could benefit from further research, but more in the vein of Esper et al., 2003 rather than this.

Finally, something I just noticed, the age-class limit found optimal in this study ($b_{max}$=270) is remarkably close to the ideal age range used in Esper at al., 2014.

*"For the final summer temperature reconstruction we only used tree rings of a certain biological age ranging from <=31 to >=306 years, i.e. removed the rings 30 years (here termed as the 'young rings') and >306 (here termed as the 'old rings') from the combined, and adjusted S88 + E12 dataset."* (Esper et al., 2014)

In the quote above S88 is the Schweingruber et al., 1988 dataset, E12 is the 587 MXD collection used in Esper et al., 2012. The similarity between the two upper limits of age class restriction further convinces me the present work provides little insight in the way contemporary Dendroclimatology applies Regional Curve Standardization.

–pjk
Stockholm

**References:**

Blackman, R.B., and Tukey, J.,W., 1958. The measurement of power spectra from the point of view of communication engineering. Dover Publications, 190 pp.

Cook, E.R. and Peters, K.: Calculating unbiased tree-ring indices for the study of climatic and environmental change, The Holocene, 7, 361–370, 1997.

Cook, E.R., Briffa, K.R., Meko, D.M., Graybill, D.A., Funkhouser, G., 1995. The segment length curse in long tree-ring chronology development for paleoclimatic studies. Holocene 5, 229e237

Esper J, Düthorn E, Krusic P, Timonen M, Büntgen U., 2014, Northern European summer temperature variations over the Common Era from integrated tree-ring density records. Journal of Quaternary Science 29, 487–494

Esper J, Frank DC, Timonen M, Zorita E, Wilson RJS, Luterbacher J, Holzkämper S, Fischer N, Wagner S, Nievergelt D, Verstege A, Büntgen U., 2012. Orbital forcing of tree-ring data. Nature Climate Change 2, 862–866.

Franke, J., Frank, D., Raible, C. C., Esper, J. & Brönnimann, S. Spectral biases in tree-ring climate proxies. Nature Clim. Change 3, 360–364 (2013).

Grudd H, Briffa KR, Karlén W et al., 2002. A 7400-year tree-ring chronologyin northern Swedish Lapland: Natural climatic variability expressed on annual to millennial timescales. The Holocene 12: 657–665.

Grudd H., 2008. Torneträsk tree-ring width and density AD 500–2004: A test

of climatic sensitivity and a new 1500-year reconstruction of north Fennoscandian summers. Climate Dynamics 31: 843–857.

Klingbjer, P. and A. Moberg, 2003. A composite monthly temperature record from Tornedalen in northern Sweden, 1802–2002, Int J Clim., 23, 1465–1494, 2003.

Koutsoyiannis, D., 2002. The Hurst phenomenon and fractional Gaussian noise made easy. Hydrol Sci J 47(4):573–595.

Mann, M.E. and J.M. Lees., 1996. Robust estimation of background noise andsignal detection in climatic time series, Clim. Change., 33, 409–445.

Melvin TM, Grudd H, Briffa K.R., 2013. Potential bias in 'updating' tree-ring chronologies using Regional Curve Standardization: re- processing the Torneträsk maximum-latewood-density data. Holocene 23: 364–373.

National Research Council (Committee on Opportunities in the Hydrologic Sciences) 1991. Opportunities in the Hydrologic Sciences. 21. National Academy Press, Washington DC, USA.

Osborn, T. J., and K.R. Briffa, 2004. Climate: The real color of climate change? Science 306, 621_622 (2004).

Park, J., 1992. Envelope estimation for quasi-periodic geophysical signals in noise: A multitaper approach, in Statistics in the Environmental and Earth Sciences, edited by A.T. Walden and P. Guttorp, 189–219, Edward Arnold, London.

Percival, D.B., and A.T. Walden, 1993. Spectral analysis for physical applications--Multitaper and conventional univariate techniques. Cambridge University, 580 pp.

Schweingruber FH, Bartholin TS, Schar E et al., 1988. Radiodensitometric dendroclimatological conifer chronologies from Lapland (Scandinavia) and the Alps (Switzerland). Boreas 17: 559-566.

Thomson, D.J., 1982. Spectrum estimation and harmonic analysis, Proc. IEEE, 70, 1055–1096.

---

## Short Comment (SC3) · 15 May 2016

In lines 32-33 the authors claim that: "In fact, Franke et al. (2013) succeeded in showing that many present day state-of-the-art reconstructions still tend to be biased in the low frequency part of the spectra."

However, more recent findings (Iliopoulou et al. 2016; Markonis and Koutsoyiannis 2016) suggest that this may not hold true. Following the approach of Bunde et al. [2013], in determining the low frequency bias by examining the long-term persistence behaviour of precipitation, it has been shown that the low frequency variability is evident in many different types of proxies and not only to the tree-rings. In addition, a simple explanation was provided for this behaviour based on the changing dependence structure of precipitation as the temporal scale increases [Markonis and Koutsoyiannis

2016].

Long-term persistence, also known as Hurst-Kolmogorov behaviour is linked to the power spectrum by a simple mathematical transformation [Beran 1994] and has been shown that is a more robust estimator of low frequency variability than the spectrum [Dimitriadis 2015].

Therefore, the authors should at least acknowledge that there is an on-going debate of whether there are low-frequency non-climatic biases in paleo records and at what extent.

Beran, J. (1994). Statistics for Long-Memory Processes, Monographs on Statistics and Applied Probability Vol. 61. Chapman and Hall, New York.

Bunde, A., Büntgen, U., Ludescher, J., Luterbacher, J., & von Storch, H. (2013). Is there memory in precipitation?. Nature Climate Change, 3(3), 174-175.

Dimitriadis, P., & Koutsoyiannis, D. (2015). Climacogram versus autocovariance and power spectrum in stochastic modelling for Markovian and Hurst–Kolmogorov processes. Stochastic Environmental Research and Risk Assessment, 29(6), 1649-1669.

Iliopoulou, T., Papalexiou, S. M., Markonis, Y., & Koutsoyiannis, D. (2016). Revisiting long-range dependence in annual precipitation. Journal of Hydrology, in press.

Markonis, Y., & Koutsoyiannis, D. (2016). Scale-dependence of persistence in precipitation records. Nature Climate Change, 6 (4), 399-401.

---

## Author Comment (AC2) · 5 Jun 2016

The manuscript touches upon an important topic within tree-ring research and climate reconstruction, namely sample replication and the ontogenetic or age trend present within tree-ring series (hence referred to as detrending methods). Especially, removing age-related trends while maintaining low-frequency climatic signals is of great inter- est to better understand earths climatic system. Both the fact that a new detrending methodology is presented to address this issue and a case study is provided on a widely used tree-ring record gives this manuscript great potential. However, the poor link to other detrending methods and work that has been done on other relevant tree- ring biases, weakens the message given by the manuscript although very strong state- ments are made. Additionally, for this method to be useful to the community more emphasis should be put on clearly explaining the steps needed to execute this de- trending technique. Because of this I would like to present a few points of discussion in relation to; a) inclusion of other detrending techniques, b) the impact of tree-ring related biases and c) expanding the methodology.

a) inclusion of other detrending techniques,

Due to both the title and the context of the paper it appears that the issues in relation to RCS deterending are present within most chronologies and has not been prop- erly addressed. However, recently a lot of work has been done on addressing and resolving detrending related issues, while these methods and sources are not being addressed or mentioned within

this manuscript. This for instance includes work on the RCS related issues in Briffa & Melvin (2011 in Hughes et al. Dendroclimatology, Developments in Paleoenvironmental Research) or the work done on comparing multi- ple dentrending methods and their potential to maintain low-frequency signals (Peters et al. 2015 Global Change Biology). Additionally other detrending methods like the signal-free detrending should be considered as they show great potential in being less affected by age-related issues (see Melvin & Briffa 2008 Dendrochornologia).

It was also surprising to see the comparison between the proposed method and the recon- struction performed by Briffa 1992, while multiple comments and revisions have been made on this chronology (see: Melvin et al. 2013 Holocene; Matskovsky et al. 2014 Climate of the Past). The current comparison is therefore very difficult to interpret as it is not clear whether one can state that one is better than the other. Additionally, the fact that the difference between the methods is very small when the sample size is higher than 14 raises more debate on sample replication than on the relevance of the method. To validate the value of the newly proposed method it is therefore crucial to include more dentrending methods and more recent Torneträsk reconstructions.

*With the aid of your comments, we will refer to articles on the newer developments. Our work is focused on the estimation of error variances and the error term studied depends only on the distribution of the measurements over the years and age classes. As far as no new trees are added to the sample, that distribution is independent of the reconstruction technique or data source. From this point of view our approach is a new one. The focus is in the error variances. It is sufficient that our reconstruction method works satisfactorily and therefore we see no need to validate our method with other methods.*

*- Explanatory text on the performance of our method will be included in the revised version of Sec. 4.3.:*

*"As far no new trees are added, the distribution of the measurements is not changed. Hence $S^2$ in Eq. (6) will be decreased but $Var_{rel}$ in Eq. (7) is unchanged. The latter term describes the error source studied here and depends only on the distribution of the measurements. Accordingly, it is more generally valid and is independent of developments in the reconstruction methods and their age dependence functions."*

*- Comparison with Briffa et al. (1992) will be rewritten in the revised version of Sec. 4.1. :*

*"In Fig. 3a the corresponding anomalies of the Torneträsk reconstruction in Briffa et al. (1992) are compared with our results. The climatologically biased quasi cycle of ca. 350 years is well seen. Especially the recent years are similarly biased showing an underestimation of the temperature. Excluding this recent underestimation, the variants of reconstructions in Melvin et al. (2013, their Fig. 4) show a similar long-term structure. Importantly, the bias is not in the tree ring measurements but in the reconstructions. Hence the bias has essentially remained as such from Briffa et al. (1992) until the present time, in spite of the developments in reconstruction methods. Naturally there can be differences in details because of different computational approaches."*

*- Fig. 4 from Melvin et al. 2013, showing different reconstructions, has been attached in the following*

   *b) the impact of tree-ring related biases*

Many biases and problem are present within tree-ring data. One of these includes the ontogenetic or age trend. However, many other biases are present like the persistent growth patterns or management related issues.

*Different Torneträsk analyses show a similar long-term variation (as far no new trees are added to the sample). Westward and eastward from Torneträsk study area there are no corresponding long-term oscillations in the nearby climatic reconstructions. Such local deviations at Torneträsk (of the order of magnitude of the greenhouse warming) are climatologically impossible and therefore the Torneträsk analyses must contain a bias.*

- *The detection of the bias will be explained in the revised version of Sec. 4.1.:*

*"In Rinne et al. 2014 it was observed that both in the nearby oceanic (August SST, Norwegian Sea, Miettinen et al. 2012) and continental (Esper et al. 2012) temperature estimates of the long term oscillations clearly and similarly differ from those derived from the Torneträsk data. Such local anomalies are climatologically impossible and therefore the mutually similar long-term oscillations in the Torneträsk reconstructions contain a bias. Accordingly, those reconstructions are suitable for our error studies.*

   *The differences observed are extreme being of the order of magnitude of the greenhouse warming. The long-term bias in*

*Torneträsk reconstructions is thus detected climatologically. In our computations we estimate the bias as the difference between the Torneträsk reconstruction and the corresponding Esper et al. (2012) reconstruction, the latter having a high number of trees"*

Multiple of these biases have been addressed in literature and have been shown to affect low-frequency patterns, which is the main interest of this manuscript. However little attention is provided on either how these biases affect the proposed method or discussed how these could affect the low-frequency signals. Work on both sampling strategy (Nerhbass-Ahles et al. 2014 Global Change Biology) and multiple biases described in literature should be addressed as these are common problems in constructing chronologies (See: Brienen et al. 2012 Global Biogeochemical Cycles; Bowman et al. 2013 Trends in Plant Science; Groenendijk et al. 2015 Global Change Biology; Autin et al. 2015 Dendrochronologia).

*The presented general theory is illustrated with the aid of the Torneträsk data. The error source introduced in our article describes the dependence on the distribution of the measurements over the years and age classes. There clearly are other error sources but they do not affect our case. The bias observed in the Torneträsk case is well explained by the error source studied, i.e. by data gaps and paucities in the data.*

*- The explanation of the bias with the aid of the error source studied will be given in the revised version of Sec. 4.2.:*

*"To illustrate the combined impact of different error terms, the degrees of freedom are computed with the aid of the approximate formula in Eq. (7) and are shown in Fig. 3c. In the beginning of the data, they grow slower than the number of measurements (Fig. 3c), show rapid changes thereafter and are often very low. Main minima and maxima of the DOF are indicated by full and open circles, respectively. Fig. 3b shows the bias estimate of the reconstruction with $b_{max}=270$. It is seen, that strong minima (maxima) of the DOF in Fig. 3c well predict following biased (unbiased) values in Fig. 3b and fit with the anomalies in reconstructions shown in Fig. 3a. "*

*c) expanding the methodology.*

For reproducibility it is essential when introducing a new method that all steps and components within the procedure are clear. In general it is therefore important that enough attention is given to the methodology section, which currently is not sufficient to apply this method on other

datasets. As an example more information should be provided on how the sensitivity analysis is performed to determine the upper limit of the age classes (bmax). Additionally, how one should determine the other required parameters to perform the computation is unclear (e.g. r1 and rn).

- *Different ways to estimate $b_{max}$ will be discussed in the last paragraph of the revised version of Sec. 4.3:*

*"There is one possibility in some specific cases to regulate the accuracy. Our method implies that an upper limit of the age classes is selected. The higher is the limit, the more measurements are included into the analysis. Simultaneously there will be more longer data gaps. It is to find a compromise between the opposite effects. An illustration is given in Fig. 1, where the cases of $b_{max}=270$ and $b_{max}=370$ are compared. In the Tornedalen case study, it was concluded to make use of $b_{max}=370$ in order to decrease the bias during recent years. That choice was motivated as the hatched line in Fig. 3a is closer to zero. Otherwise it is seen that the changes are weak and the selection of $b_{max}$ is not decisive. If it wished to be made mechanically, a possibility would be to minimize $Var_{ave}$ with respect to $b_{max}$ in Eq. (7). In Esper et al. (2014), the upper limit was taken to be $b_{max}=306$."*

- *Determination of $r_1$ and $r_n$ will be explicitly described in Sec. 2.2.:*

*"In order to get the average taken over all age classes, the contributions of the missing young and old age classes are needed. These can be estimated as follows. First estimates of measurements are interpolated for every age class where that is possible. Then years with values in age classes of $b=20$ and $b=270$ are selected. This makes it possible to compute yearly averages over all age classes of 20 thru 270. Next yearly averages over age classes of 21 thru 270 are computed. Generally this new average is smaller but the variation between the years is strong. By computing the mean and r.m.s.e. of the yearly differences between the averages with and without $b=20$ we get an estimate of the eliminated age class $b=20$. In the next step the yearly averages over age classes of 22 thru 270 are computed and used to estimate the impact of missing age classes of $b=21$ and $b=22$ to the average of all measurements between $b=20$ and $b=270$. The computations are continued by dropping*

*out more age classes. Similar approach is applied to estimate the impact of missing old age classes.*

*The estimation turned out to be more complicated if only very young age classes ($b1 < \approx 20$) or most of the older age classes ($b_n < 30$) were missing. To keep the formulae simple, the linear approximation is extended to those cases, too.*

*The result is that the contributions of the missing young and old age classes to the average taken over all age classes can be estimated linearly by $M_{young} \approx r_1(b_1-1)$ and $M_{old} \approx r_n(b_n-b_{max})$, respectively, where $r_1 \approx 0.000128$ and $r_n \approx 0.000170$. Note that $M_{old}$ results in a negative correction. The accuracy of such corrections turned out to be low their variances being roughly $[r_1(b_1-1)]^2$ and $[r_n(b_n-b_{max})]^2$. The application of the corrections are illustrated for an individual year in Fig. 2 for $b_{max}=300$."*

*The determination of $r_1$ and $r_n$ can be varied by applying different age classes. In any case the error variances are high and the values of $r_1$ and $r_n$ cannot be determined accurately in the presence of data gaps and paucities.*

What is also vague within this method is how the extrapolation affects the results (see Figure 2). If there is for instance a year with only a few young age classes a large proportion of the mean is determined by the extrapolated older classes, which are heavily dependent upon the assumption you make within this extrapolation procedure (which, if I understand correctly, in the manuscript is proposed as a negative linear relationship which can be highly debated). How these situations affect the method should be described or analysed.

*The impact of the missing younger and older age classes is described by the error variances. In the case of data gaps and paucities, the error variances become high and the reconstruction will be less reliable in these instances*

Finally, the error estimation and the representation of missing age classes in Figure 3 could be of great value to the community. Especially, visually showing where specific age-classes are lacking could help to detect specific biases and to disentangle whether low-frequency signals are caused by sample replication or climate. Because of this relevance I feel more attention should be given to the methods section on this as currently the description is highly condensed and in some parts not clear.

*We wish that the clarifications and explanations to be added into the manuscript make the text clearer.*
* * *
*Attachement*

Fig. 4 in Melvin et al. 2013 has been referred to in the manuscript. It is attached here (https://crudata.uea.ac.uk/cru/papers/melvin2012holocene/).

There are several reconstructions from several years (e.g. Briffa MXD 1992 red+green in the uppermost panel, Grudd TRW 2002 blue in the next one, Briffa TRW 2011 blue; recent ones in the three lower panels). All of them show similar long-term oscillations that are seen in Briffa 1992 MXD. It is seen that recent advanced reconstruction methods have not changed the situation.

Exceptions are only seen during recent years where the "divergence" problem is best seen in Briffa 1992 MXD (green line). There are several attempts to correct the problem. No similar variations in the "divergence" case (systematic underestimation of the temperature) after 1600 AD are visible. The cold bias is similarly in different reconstructions of the order of magnitude of the greenhouse warming.

[Figure]

**Changes to the manuscript.**

Contents

*2.2 Explicit computation formulae*

> *The estimation of parameters $r_1$ and $r_n$ is described explicitly*

*3 Torneträsk case study*

> *The original and corrected observations for 1826-38 are presented in Fig. 4. The motivation of the correction is presented in details.*

*4.1. Bias of the Torneträsk reconstructions*

> *The climatological detection of the bias in the Torneträsk*

*reconstructions is presented more specifically.*

*The use of Briffa (1992) reconstruction in Fig. 3a is explained in more detail.*

**4.2. Explanation of the bias observed**

*It is pointed out that the Torneträsk case is only an illustration and application of the general error terms in Eqs. (4) and (5).*

*The bias in the reconstructions is, on the basis of the description in Sec. 4.1, estimated with the aid of the reconstruction in Esper et al. (2012).*

**4.3 The performance of the reconstruction method applied**

*The performance of the reconstruction is described only to show that the error analysis is based on a sound calculation. The resulting temperature estimates are sufficiently satisfactory.*

*It is explained that the reconstruction methods published in the literature do not impact the error source studied here and are thus outside of the scope of our study.*

*The selection of the upper limit of the age classes is discussed.*

**2.2 Explicit computation formulae**

*(page 4, lines 9-23, the older version of the following new paragraph began with "Assume ...")*

In order to get the average taken over all age classes, the contributions of the missing young and old age classes are needed. These can be estimated as follows. First estimates of measurements are interpolated for every age class where that is possible. Then years with values in age classes of $b=20$ and $b=270$ are selected. This makes it possible to compute yearly averages over all age classes of 20 thru 270. Next yearly averages over age classes of 21 thru 270 are computed. Generally this new average is smaller but the variation between the years is strong. By computing the mean and r.m.s.e. of the yearly differences between the averages with and without b=20 we get an estimate of the eliminated age class $b=20$. In the next step the yearly averages over age classes of 22 thru 270 are

computed and used to estimate the impact of missing age classes of $b=21$ and $b=22$ to the average of all measurements between $b=20$ and $b=270$. The computations are continued by dropping out more age classes. Similar approach is applied to estimate the impact of missing old age classes.

The estimation turned out to be more complicated if only very young age classes ($b_1 <\approx 20$) or most of the older age classes ($b_n < 30$) were missing. To keep the formulae simple, the linear approximation is extended to those cases, too.

The result is that the contributions of the missing young and old age classes to the average taken over all age classes can be estimated linearly by $M_{young} \approx r_1(b_1-1)$ and $M_{old} \approx r_n(b_n - b_{max})$, respectively, where $r_1 \approx 0.000128$ and $r_n \approx 0.000170$. Note that $M_{old}$ results in a negative correction. The accuracy of such corrections turned out to be low their variances being roughly $[r_1(b_1-1)]^2$ and $[r_n(b_n-b_{max})]^2$. The application of the corrections are illustrated for an individual year in Fig. 2 for $b_{max} = 300$.

The focus here is in the long-term error. Important are the estimates of the error variances due to the missing age classes. Precise values of $r_1$ and $r_n$ are not needed. The main information included in the results is the dependence of the variance terms on the lengths of the data gaps, $b_1-1$ and $b_{max}-b_n$. Our reconstruction method is directed to uncover that dependence explicitly in a simple way. Otherwise it is enough that the reconstruction method performs sufficiently satisfactorily. It needs not be an optimal one.

No assumption on the age function is made. The contributions of the young and old age classes can widely vary between years as indicated by their error variances. However, average corrections are needed in order to decrease the impact of systematic errors. By adding the corrections, the yearly average of the interpolated, extrapolated and measured values becomes approximately **…**

**3 Torneträsk case study**

*(page 7, lines 1-6, the changes are given in blue)*

Klingberj and Moberg (2003) composited a series of instrumental observations made in the Tornedalen region, some 300 km from Torneträsk. Their construction begins with instrumental data from

Övertorneå (1802-1838). The observation hours were not stated explicitly for 1826-38 and the authors had to assume them. The JJA temperatures are known to be sensitive to the observation hours used. If these are not precisely known, true climatic variations may remain but the average level of the temperature may become wrong. In such cases any kind of support is welcome. During 1826-38 the bias estimate in our reconstructions is rather small (≈ -0.25°C, Fig. 3a) and the DOF are rather high (≈11, Fig. 3c). Our reconstruction can therefore be applicable. If the JJA temperatures in observations are systematically increased by 1°C every year during 1826-38, the smoothed result happens to fit rather well with our reconstruction in Fig. 4. Accordingly, the corrected temperatures are supported by the reconstruction and the anomalous coldness in observations before 1840 in Fig. 4 seems suspicious due to the unknown observation hours. The comparison with instrumental data here - while interesting - is not essential, moreover as in any case temperatures before ca. 1850 may contain biases (Melvin et al., 2013; Grudd, 2008).

*(corrected and original observations are now shown in Fig.4)*

[Figure]

Figure 4. Comparison of Torneträsk temperature reconstruction and Tornedalen temperature observations 1802-1977. Reconstruction mean has been adjusted to fit the corresponding observational mean during 1850-1950 and both curves have been smoothed.
* * *
*(revised version of the discussion, page 7, line 17 – page 9, line 7)*

**4.1. Bias of the Torneträsk reconstructions**

In Rinne et al. 2014 it was observed that both in the nearby oceanic (August SST, Norwegian Sea, Miettinen et al. 2012) and continental (Esper et al. 2012) temperature estimates of the long term oscillations clearly and similarly differ from those derived from the Torneträsk data. Such local anomalies are climatologically impossible and therefore the mutually similar long-term oscillations in the Torneträsk reconstructions contain a bias. Accordingly, those reconstructions are suitable for our error studies.

The differences observed are extreme being of the order of magnitude of the greenhouse warming. The long-term bias in Torneträsk reconstructions is thus detected climatologically. In our computations we estimate that climatological bias as the difference between the Torneträsk reconstruction and the corresponding Esper et al. (2012) reconstruction, the latter having a high number of trees. In Fig. 3a the corresponding anomalies of the Torneträsk reconstruction in Briffa et al. (1992) are compared with our results. The climatologically biased quasi cycle of ca. 350 years is well seen. Especially the recent years are similarly biased showing an underestimation of the temperature. Excluding this recent underestimation, the variants of reconstructions in Melvin et al. (2013, their Fig. 4) show a similar long-term structure. Importantly, the bias is not in the tree ring measurements but in the reconstructions. Hence the bias has essentially remained as such from Briffa et al. (1992) until the present time, in spite of the developments in reconstruction methods. Naturally there can be differences in details because of different computational approaches.

**4.2. Explanation of the bias observed**

The error formulae derived are of a general form (Eqs. 4 and 5).

The error variances depend on the relative error terms and the dimensional portions $(b_{max}\, r_1)^2$, $(b_{max}\, r_n)^2$ and $S^2$. Because the relative error terms depend only on the distribution of the measurements over the age classes and years, they can be applied to any data without knowing the measurements or reconstruction method used.

In Fig. 3 the relative variances are applied for the Torneträsk data with 65 trees. Paucities in the tree population or periods of missing trees (Fig. 1) cause successively data gaps in young, intermediate and old age classes leading to variance variations in Fig. 3d. The increased variance decrease the degrees of freedom and so makes the bias more probable, which is then seen as erroneous long-term oscillations in the reconstructions.

To illustrate the combined impact of different error terms, the degrees of freedom are computed with the aid of the approximate formula in Eq. (7) and are shown in Fig. 3c. In the beginning of the data, they grow slower than the number of measurements (Fig. 3c), show rapid changes thereafter and are often very low. Main minima and maxima of the DOF are indicated by full and open circles, respectively. Fig. 3b shows the bias estimate of the reconstruction with $b_{max}$=270. It is seen, that strong minima (maxima) of the DOF in Fig. 3c well predict following biased (unbiased) values in Fig. 3b and fit with the anomalies in reconstructions shown in Fig. 3a.

Low values of the DOF ($<5$) are related to extrema of the bias. High values of the DOF ($>14$) indicate vanishing bias. The latter ones are not numerous and therefore bias-free cases are seen only temporarily in Fig. 3b.

One year with low DOF is separately indicated in Fig. 3. A change in the tree population is seen in 1356 AD (Fig. 1) where the oldest age class ($<270$ years) drops suddenly down to b=116 years. As a consequence, variances of the older and intermediate age classes are peaked in Fig. 3d, further lowering strongly the DOF in Fig. 3c. This is reflected in the bias (Fig. 3b) and in the reconstruction (Fig. 3a, smoothed values in Figs. 3a and 3b naturally lag the sudden changes in Figs. 3c and 3d). Correspondingly the optimal distribution of the measurements around 869 and 1500 AD (Fig. 1) are reflected in low error variances (Fig. 3d), high number of the DOF (Fig. 3c) and unbiased temperature estimates (Fig. 3b).

In Fig. 1, there is a repetitive similarity between the two longer periods with nearly continuous samples of trees (AD 1250-1500 and

1600-1800) and two periods without new trees (AD 1500-1600 and 1800-1980). The corresponding terms of the relative variance (Eq. 5) will be discussed in the following.

The large first peaks of variance around 1356 AD (Fig. 3d) result from missing intermediate and old age classes. These damp slowly out and new peaks are seen at 1600 AD. Here they are due to the intermediate (interpolated) and young age classes. The same structure is repeated between 1600-1800 AD and 1800-1980 AD. In this way the paucities due to missing trees in the data cause alternating data gaps leading to varying behavior in temperature reconstructions. Especially biases around 1600 and 1950 can be explained by the error terms connected to the missing younger age classes (due to the missing trees). The latter case is known as "divergence", a systematic underestimation of the temperatures (Briffa et al. 1998, D'Arrigo et al. 2008). As there is no formal difference between the cases of 1600 and 1950, it is natural to refer to "divergence" in both cases.

**4.3 The performance of the reconstruction method applied**

The reconstruction method has been designed for estimation of the error variances. A general form of variances is found in Eq. (5). This is applicable as well for tree ring widths as maximum latewood densities. Only three parameters are needed in explicit applications ($r_1$, $S$, $r_n$). Error sources are not corrected but retained in order to study them. For instance, the long-term erroneous oscillations in Briffa 1992 are well reproduced in Fig. 3a during recent years, too. As a summary, the reconstruction method performs as it should.

The reconstruction method gives two values for every year, an average and its error estimate. If the parameters ($r_1$, $S$, $r_n$) were known, yearly confidence limits of the reconstruction could be given. An important part here is the variances due to missing young and old age classes.

To illustrate, the Torneträsk data is applied. Figs. 3a and 4 show that the reconstruction method performs sufficiently satisfactorily and so the error estimates can be seen to be reasonable. On the other hand, the relative errors due to data gaps and paucities are seldom low (e.g. years 869 and 1500 in Fig. 3d). Therefore the reconstruction can be expected to be inaccurate during most years.

If the errors due to the distribution of the measurements are not taken into account, in Eq. (6) $Var_{rel}=1$. Conventionally the sample

accuracy is then characterized by presenting the yearly number of measurements. This practise is followed here. Instead of trying to estimate confidence limits, the sample accuracy is characterized by giving the degrees of freedom (Fig. 3c). The practical application requires that the general error terms in Eq. (5) are approximated and compressed into Eqs. (6) and (7).

In the recent literature new methods to estimate the age dependence function more precisely are introduced (e.g. Briffa et al. 2013; Melvin et al. 2013; Matskovsky and Helama 2014). They improve the reconstruction and the errors in the long-term oscillations may be decreased. Potential error sources in the sampling technique have been detected (e.g. Bowman et al. 2013). As far no new trees are added, the distribution of the measurements is not changed. Hence $S^2$ in Eq. (6) will be decreased but $Var_{rel}$ in Eq. (7) is unchanged. The latter term describes the error source studied here and depends only on the distribution of the measurements. Accordingly, it is more generally valid and is independent of developments in the reconstruction methods and their age dependence functions.

In our reconstruction method the missing younger and older age classes are taken into account. It is natural that the results resemble to those that use age dependence functions. However, here the impact of the missing younger and older age classes is estimated directly from the measurements and only for the average impact of the missing age classes. The resulting estimates of error variances are high in the cases of data gaps and paucities. Therefore there is here no need for a more advanced reconstruction method because necessarily the error estimates will be high and the estimated impact of missing age classes less accurate.

There is one possibility in some specific cases to regulate the accuracy. Our method implies that an upper limit of the age classes is selected. The higher is the limit, the more measurements are included into the analysis. Simultaneously there will be more longer data gaps. It is to find a compromise between the opposite effects. An illustration is given in Fig. 1, where the cases of $b_{max}=270$ and $b_{max}=370$ are compared. In the Tornedalen case study, it was concluded to make use of $b_{max}=370$ in order to decrease the bias during recent years. That choice was motivated as the hatched line in Fig. 3a is closer to zero. Otherwise it is seen that the changes are weak and the selection of $b_{max}$ is not decisive. If it wished to be made

mechanically, a possibility would be to minimize $\text{Var}_{ave}$ with respect to $b_{max}$ in Eq. (7). In Esper et al. (2014), the upper limit was taken to be $b_{max}=306$.

---

## Author Comment (AC3) · 5 Jun 2016

This paper introduces a "new" standardisation method to construct chronologies. It tries to show that there is low-frequency bias in tree ring chronology reconstructions. The new reconstruction method is used with the Tornetrask MXD data from 1988 to demonstrate this bias.

The authors state that "The presented method to estimate past temperatures from tree ring measurements is a new approach, where no age dependence of the tree rings is estimated." yet they are clearly removing the age related growth of trees as a linear trend. For each calendar year they have a few measured rings (e.g. 20 values). They effectively create further values by extrapolation including younger rings and older rings to a total of 270 (e.g. adding 250 values) with a linear age-related decay. These 270 values are then averaged together thus removing the effect of ring-width aging using the presumption of some form of liner decay of ring width with age.

The method of estimating R1 and Rn (page 4 line 17) will need a detailed explanation.

*- Determination of $r_1$ and $r_n$ is now explicitly described in the revised version of Sec. 2.2.*

Overall it is likely to have a similar result to that of creating and fitting a linearly decaying RCS curve. In RCS the averaging and smoothing of the RCS curve tends to reduce the climate noise from the estimation of the ageing trend whereas in the proposed method using rings from a single

year which all have the same climate signal achieves this. Their conclusion that they do not remove the effect of age-related growth from their measurements is not justified as they do try to remove the age effect.

*Our method is designed to estimate error variances. Especially important are the variances of the systematic errors due to the missing young and old age classes. It is natural that in their estimation the reconstruction method resembles usual conventions. However, here the impact of the missing younger and older age classes is estimated directly from the measurements and only for the average impact of the missing young and old age classes. Otherwise the trapezoidal rule is applied.*

*- Explanatory text on the performance of our method is included in the revised version of Sec. 4.3.*

The 1988 Tornetrask MXD data were selected by age (oldest well behaved trees) from the much larger TRW data set with a view to using curve-fitting standardisation meth- ods with sufficient replication for reconstructing medium frequency variability. An even distribution of tree rings by age in each year was not thought necessary in 1988. This is not a suitable data set to introduce or evaluate the proposed new standardisation technique.

*It is true that there were better data for such an evaluation if that were needed. From our point of view, the 1988 Torneträsk is suitable to illustrate the performance of the error formulae derived. Otherwise it is enough that our reconstruction method performs suffeciently well (the performance is illustrated with the aid of the Torneträsk data).*

*- Comparison with Briffa et al. (1992) will be reconsidered in Sec. 4.1.*

The authors need to note that Briffa et al (2009, Hughes book chapter) show that for these MXD measurements from Tornetrask the assumption of linear decay creates a bias in the reconstruction. For TRW, the assumption of linear decay would create even more bias.

*The estimates of the error variance are so high that in any case the confidence limits of our reconstruction were wide if such limits had been constructed. The linear decay is not asssumed but estimated to be applicable in missing young and old age classes. It follows that there are two results of the estimation: the average impact of those age classes and the corresponding error variance.*

*- Confidence limits are implicitly chracterized by the degrees of freedom and are discussed in the revised version of Sec. 4.3.*

The Esper 2012 chronology has more trees (even after using mean tree rather than multiple cores) and less error due to sample count i.e. less noise. Is the age distribution of the Esper trees biased over time? No assessment is made of this so the presence or absence of systematic bias is not known and the comparison (and conclusions based on it) in this paper is not justified.

There is no attempt to distinguish between bias due to the age-related growth decay in tree measurements and noise created by poor replication and the authors confuse these two effects making their conclusions less valid and unhelpful.

*The detection of the bias is now explained more in details, in the manuscript only a reference was given. Different Torneträsk analyses show a similar long-term variation (as far no new trees are added to the sample). Westward and eastward from Torneträsk there are no corresponding long-term oscillations in the nearby reconstructions. Such local deviations at Torneträsk (of the order of magnitude of the greenhouse warming) are climatologically impossible and therefore the Torneträsk analyses must be biased.*

*- The detection of the bias will be explained in Sec. 4.1 of the revised manuscript as follows.:*

> *"In Rinne et al. 2014 it was observed that both in the nearby oceanic (August SST, Norwegian Sea, Miettinen et al. 2012) and continental (Esper et al. 2012) temperature estimates of the long term oscillations clearly and similarly differ from those derived from the Torneträsk data. Such local anomalies are climatologically impossible and therefore the mutually similar long-term oscillations in the Torneträsk reconstructions contain a bias. Accordingly, such reconstructions are suitable for our error studies.*

> *The differences observed are extreme being of the order of magnitude of the greenhouse warming. The long-term bias in Torneträsk reconstructions is thus detected climatologically. In our computations we estimate that climatological bias as the difference between the Torneträsk reconstruction and the corresponding Esper et al. (2012) reconstruction, the latter having a high number of trees"*

*- The explanation of the bias with the aid of the error source studied is given in the revised version of Sec. 4.2.*

The presentation in this paper is not suitable to introduce a "new" standardisation method. A comparison of new against existing methods is needed which shoulfd include a careful assessment of errors – with separation of noise related to insufficient samples and systematic bias related to poor removal of age-dependent growth and an evaluation of error magnitude.

*Our estimate of the error variance is divided in two parts which describe the error sources that you mentioned:  the noise related to insufficient samples and  systematic bias related to poor removal of age-dependent growth. The former part is studied in the article. The latter one is widely studied in the literature and is outside of the scope of our study.*

A sample data set with sufficient samples in each year to sub-divide the data and show the effect of reducing sample counts is needed and only then can the bias due to age-trend be shown.

*Our work is focused on the estimation of error variances and the error term studied depends only on the distribution of the measurements over the years and age classes. Our approach is designed to estimate the variance of that error. Otherwise it is enough  that our reconstruction works sufficiently satisfactory and therefoer there is no need to validate our method with other methods.*

*- Explanatory text on the performance of our method is included in the revised version of Sec. 4.3.*

My overall assessment is that this paper requires considerable improvement before it is suitable for publication.

*We wish that the clarifications and explanations made in the manuscript have make the text clearer*

Dr Thomas Melvin
**Changes to the manuscript.**

*Contents*

*2.2 Explicit computation formulae*

*The estimation of parameters $r_1$ and $r_n$ is described explicitly*

*3 Torneträsk case study*

*The original and corrected observations for 1826-38 are presented in Fig. 4. The motivation of the correction is presented in details.*

*4.1. Bias of the Torneträsk reconstructions*

*The climatological detection of the bias in the Torneträsk reconstructions is presented more specifically.*
*The use of Briffa (1992) reconstruction in Fig. 3a is explained in more detail.*

*4.2. Explanation of the bias observed*

*It is pointed out that the Torneträsk case is only an illustration and application of the general error terms in Eqs. (4) and (5).*
*The bias in the reconstructions is, on the basis of the description in Sec. 4.1, estimated with the aid of the reconstruction in Esper et al. (2012).*

*4.3 The performance of the reconstruction method applied*

*The performance of the reconstruction is described only to show that the error analysis is based on a sound calculation. The resulting temperature estimates are sufficiently satisfactory.*
*It is explained that the reconstruction methods published in the literature do not impact the error source studied here and are thus outside of the scope of our study.*
*The selection of the upper limit of the age classes is discussed.*

**2.2 Explicit computation formulae**

*(page 4, lines 9-23, the older version of the following new paragraph began with "Assume ...")*

In order to get the average taken over all age classes, the contributions of the missing young and old age classes are needed. These can be estimated as follows. First estimates of measurements are interpolated for every age class where that is possible. Then years with values in age classes of $b=20$ and $b=270$ are selected. This makes it possible to compute yearly averages over all age classes of 20 thru 270. Next yearly averages over age classes of 21 thru 270 are computed. Generally this new average is smaller but the variation between the years is strong. By computing the mean and r.m.s.e. of the yearly differences between the averages with and without b=20 we get an estimate of the eliminated age class $b=20$. In the next step the yearly averages over age classes of 22 thru 270 are computed and used to estimate the impact of missing age classes of $b=21$ and $b=22$ to the average of all measurements between $b=20$ and $b=270$. The computations are continued by dropping out more age classes. Similar approach is applied to estimate the impact of missing old age classes.

The estimation turned out to be more complicated if only very young age classes ($b_1<\approx 20$) or most of the older age classes ($b_n<30$) were missing. To keep the formulae simple, the linear approximation is extended to those cases, too.

The result is that the contributions of the missing young and old age classes to the average taken over all age classes can be estimated linearly by $M_{young}\approx r_1(b_1-1)$ and $M_{old}\approx rn(b_n-b_{max})$, respectively, where $r_1\approx 0.000128$ and $r_n\approx 0.000170$. Note that $M_{old}$ results in a negative correction. The accuracy of such corrections turned out to be low their variances being roughly $[r_1(b_1-1)]^2$ and $[rn(b_n-b_{max})]^2$. The application of the corrections are illustrated for an individual year in Fig. 2 for $b_{max}=300$.

The focus here is in the long-term error. Important are the estimates of the error variances due to the missing age classes. Precise values of $r_1$ and $r_n$ are not needed. The main information included in the results is the dependence of the variance terms

on the lengths of the data gaps, $b_1$-1 and $b_{max}$-$b_n$. Our reconstruction method is directed to uncover that dependence explicitly in a simple way. Otherwise it is enough that the reconstruction method performs sufficiently satisfactorily. It needs not be an optimal one.

No assumption on the age function is made. The contributions of the young and old age classes can widely vary between years as indicated by their error variances. However, average corrections are needed in order to decrease the impact of systematic errors. By adding the corrections, the yearly average of the interpolated, extrapolated and measured values becomes approximately **…**

**3 Torneträsk case study**

*(page 7, lines 1-6, the changes are given in* blue*)*

Klingberj and Moberg (2003) composited a series of instrumental observations made in the Tornedalen region, some 300 km from Torneträsk. Their construction begins with instrumental data from Övertorneå (1802-1838). The observation hours were not stated explicitly for 1826-38 and the authors had to assume them. The JJA temperatures are known to be sensitive to the observation hours used. If these are not precisely known, true climatic variations may remain but the average level of the temperature may become wrong. In such cases any kind of support is welcome. During 1826-38 the bias estimate in our reconstructions is rather small (≈ -0.25°C, Fig. 3a) and the DOF are rather high (≈11, Fig. 3c). Our reconstruction can therefore be applicable. If the JJA temperatures in observations are systematically increased by 1°C every year during 1826-38, the smoothed result happens to fit rather well with our reconstruction in Fig. 4. Accordingly, the corrected temperatures are supported by the reconstruction and the anomalous coldness in observations before 1840 in Fig. 4 seems suspicious due to the unknown observation hours. The comparison with instrumental data here - while interesting - is not essential, moreover as in any case temperatures before ca. 1850 may contain biases (Melvin et al., 2013; Grudd, 2008).

*(corrected and original observations are now shown in Fig.4)*

[Figure]

Figure 4. Comparison of Torneträsk temperature reconstruction and Tornedalen temperature observations 1802-1977. Reconstruction mean has been adjusted to fit the corresponding observational mean during 1850-1950 and both curves have been smoothed.
* * *
*(revised version of the discussion, page 7, line 17 – page 9, line 7)*

**4.1. Bias of the Torneträsk reconstructions**

In Rinne et al. 2014 it was observed that both in the nearby oceanic (August SST, Norwegian Sea, Miettinen et al. 2012) and continental (Esper et al. 2012) temperature estimates of the long term oscillations clearly and similarly differ from those derived from the Torneträsk data. Such local anomalies are climatologically impossible and therefore the

mutually similar long-term oscillations in Torneträsk reconstructions contain a bias. Accordingly, those reconstructions are suitable for our error studies.

The differences observed are extreme being of the order of magnitude of the greenhouse warming. The long-term bias in Torneträsk reconstructions is thus detected climatologically. In our computations we estimate that climatological bias as the difference between the Torneträsk reconstruction and the corresponding Esper et al. (2012) reconstruction, the latter having a high number of trees. In Fig. 3a the corresponding anomalies of the Torneträsk reconstruction in Briffa et al. (1992) are compared with our results. The climatologically biased quasi cycle of ca. 350 years is well seen. Especially the recent years are similarly biased showing an underestimation of the temperature. Excluding this recent underestimation, the variants of reconstructions in Melvin et al. (2013, their Fig. 4) show a similar long-term structure. Importantly, the bias is not in the tree ring measurements but in the reconstructions. Hence the bias has essentially remained as such from Briffa et al. (1992) until the present time, in spite of the developments in reconstruction methods. Naturally there can be differences in details because of different computational approaches.

**4.2. Explanation of the bias observed**

The error formulae derived are of a general form (Eqs. 4 and 5). The error variances depend on the relative error terms and the dimensional portions $(b_{max} r_1)^2$, $(b_{max} r_n)^2$ and $S^2$. Because the relative error terms depend only on the distribution of the measurements over the age classes and years, they can be applied to any data without knowing the measurements or reconstruction method used.

In Fig. 3 the relative variances are applied for the Torneträsk data with 65 trees. Paucities in the tree population or periods of missing trees (Fig. 1) cause successively data gaps in young, intermediate and old age classes leading to variance variations in Fig. 3d. The increased variance decrease the degrees of freedom and so makes the bias more probable, which is then seen as erroneous long-term oscillations in the reconstructions.

To illustrate the combined impact of different error terms, the degrees of freedom are computed with the aid of the approximate formula in Eq. (7) and are shown in Fig. 3c. In the beginning of the data, they grow slower than the number of measurements (Fig. 3c), show rapid changes thereafter and are often very low. Main minima and maxima of

the DOF are indicated by full and open circles, respectively. Fig. 3b shows the bias estimate of the reconstruction with $b_{max}$=270. It is seen, that strong minima (maxima) of the DOF in Fig. 3c well predict following biased (unbiased) values in Fig. 3b and fit with the anomalies in reconstructions shown in Fig. 3a.

Low values of the DOF (<5) are related to extrema of the bias. High values of the DOF (>14) indicate vanishing bias. The latter ones are not numerous and therefore bias-free cases are seen only temporarily in Fig. 3b.

One year with low DOF is separately indicated in Fig. 3. A change in the tree population is seen in 1356 AD (Fig. 1) where the oldest age class (<270 years) drops suddenly down to b=116 years. As a consequence, variances of the older and intermediate age classes are peaked in Fig. 3d, further lowering strongly the DOF in Fig. 3c. This is reflected in the bias (Fig. 3b) and in the reconstruction (Fig. 3a, smoothed values in Figs. 3a and 3b naturally lag the sudden changes in Figs. 3c and 3d). Correspondingly the optimal distribution of the measurements around 869 and 1500 AD (Fig. 1) are reflected in low error variances (Fig. 3d), high number of the DOF (Fig. 3c) and unbiased temperature estimates (Fig. 3b).

In Fig. 1, there is a repetitive similarity between the two longer periods with nearly continuous samples of trees (AD 1250-1500 and 1600-1800) and two periods without new trees (AD 1500-1600 and 1800-1980). The corresponding terms of the relative variance (Eq. 5) will be discussed in the following.

The large first peaks of variance around 1356 AD (Fig. 3d) result from missing intermediate and old age classes. These damp slowly out and new peaks are seen at 1600 AD. Here they are due to the intermediate (interpolated) and young age classes. The same structure is repeated between 1600-1800 AD and 1800-1980 AD. In this way the paucities due to missing trees in the data cause alternating data gaps leading to varying behavior in temperature reconstructions. Especially biases around 1600 and 1950 can be explained by the error terms connected to the missing younger age classes (due to the missing trees). The latter case is known as "divergence", a systematic underestimation of the temperatures (Briffa et al. 1998, D'Arrigo et al. 2008). As there is no formal difference between the cases of 1600 and 1950, it is natural to refer to "divergence" in both cases.

**4.3 The performance of the reconstruction method applied**

The reconstruction method has been designed for estimation of the error variances. A general form of variances is found in Eq. (5). This is applicable as well for tree ring widths as maximum latewood densities. Only three parameters are needed in explicit applications ($r_1$, $S$, $r_n$). Error sources are not corrected but retained in order to study them. For instance, the long-term erroneous oscillations in Briffa 1992 are well reproduced in Fig. 3a during recent years, too. As a summary, the reconstruction method performs as it should.

The reconstruction method gives two values for every year, an average and its error estimate. If the parameters ($r_1$, $S$, $r_n$) were known, yearly confidence limits of the reconstruction could be given. An important part here is the variances due to missing young and old age classes.

To illustrate, the Torneträsk data is applied. Figs. 3a and 4 show that the reconstruction method performs sufficiently satisfactorily and so the error estimates can be seen to be reasonable. On the other hand, the relative errors due to data gaps and paucities are seldom low (e.g. years 869 and 1500 in Fig. 3d). Therefore the reconstruction can be expected to be inaccurate during most years.

If the errors due to the distribution of the measurements are not taken into account, in Eq. (6) $\text{Var}_{rel}=1$. Conventionally the sample accuracy is then characterized by presenting the yearly number of measurements. This practise is followed here. Instead of trying to estimate confidence limits, the sample accuracy is characterized by giving the degrees of freedom (Fig. 3c). The practical application requires that the general error terms in Eq. (5) are approximated and compressed into Eqs. (6) and (7).

In the recent literature new methods to estimate the age dependence function more precisely are introduced (e.g. Briffa et al. 2013; Melvin et al. 2013; Matskovsky and Helama 2014). They improve the reconstruction and the errors in the long-term oscillations may be decreased. Potential error sources in the sampling technique have been detected (e.g. Bowman et al. 2013). As far no new trees are added, the distribution of the measurements is not changed. Hence $S^2$ in Eq. (6) will be decreased but $Var_{rel}$ in Eq. (7) is unchanged. The latter term describes the error source studied here and depends only on the

distribution of the measurements. Accordingly, it is more generally valid and is independent of developments in the reconstruction methods and their age dependence functions.

In our reconstruction method the missing younger and older age classes are taken into account. It is natural that the results resemble to those that use age dependence functions. However, here the impact of the missing younger and older age classes is estimated directly from the measurements and only for the average impact of the missing age classes. The resulting estimates of error variances are high in the cases of data gaps and paucities. Therefore there is here no need for a more advanced reconstruction method because necessarily the error estimates will be high and the estimated impact of missing age classes less accurate.

There is one possibility in some specific cases to regulate the accuracy. Our method implies that an upper limit of the age classes is selected. The higher is the limit, the more measurements are included into the analysis. Simultaneously there will be more longer data gaps. It is to find a compromise between the opposite effects. An illustration is given in Fig. 1, where the cases of $b_{max}=270$ and $b_{max}=370$ are compared. In the Tornedalen case study, it was concluded to make use of $b_{max}=370$ in order to decrease the bias during recent years. That choice was motivated as the hatched line in Fig. 3a is closer to zero. Otherwise it is seen that the changes are weak and the selection of $b_{max}$ is not decisive. If it wished to be made mechanically, a possibility would be to minimize $Var_{ave}$ with respect to $b_{max}$ in Eq. (7). In Esper et al. (2014), the upper limit was taken to be $b_{max}=306$.

---

## Author Comment (AC4) · 5 Jun 2016

*(Following the comments presented by the referees, we add much explanation and clarification into Secs. 2.2,3,4.1,4.2 and 4.3 of our manuscript. We apologize that this has taken much time and our answer is delayed. The revised sections have been attached.)*

*(Our response is in the following given in italics)*

Clim. Past Discuss., doi:10.5194/cp–2016–27, 2016 Manuscript under review for journal Clim. Past Published: 23 March 2016

**A universal error source in past climate estimates derived from treerings**

Juhani Rinne[1], Mikko Alestalo[1] and Jörg Franke[2,3]

[1]Finnish Meteorological Institute, P. O. Box 503, FI – 00101 Helsinki, Finland [2]Institute of Geography, University of Bern, 3012 5 Bern, Switzerland [3]Oeschger Centre for Climate Change Research, University of Bern, 3012

Bern, SwitzerlandCorrespondence to: H. J. Rinne (juhani.rinne@kolumbus.fi)

**Article Review:** pjk–Stockholm, SE

**Introduction:**

In this study (hereafter: RN()) the authors start out to prove that paucities and age–class gaps in the Torneträsk MXD chronology (Schweingruber 1988, Briffa et al., 1992) are responsible for a low frequency (red) bias. To prove this RN() have developed a method of MXD chronology construction that purportedly accounts for paucities and age–class gaps. They evidence their method's advantage by comparing a reconstruction of the historical Tornedalen temperature record (Klingbjer and Moberg, 2003) to a reconstruction produced by their method. As evidence for the low frequency bias in the RCS chronology produced by Briffa et al.,

1992 (hereafter: BF92), the authors rely on a simple visual comparison.

It is clear that much of the motivation for this study relies on the theories proposed in Franke et al., 2013; hereafter: FK2013. One cannot fail to get this message from the rather brazen title

*We have changed the title to be less bold.*

and opening sentence of the Abstract (which requires references),

*We have understood that it is not recommendable to use references in the abstract especially here where FK2013 is presented in the introduction.*

and the last sentence (for which the study provides no evidence).

*The divergence cases are explicitly discussed in the connection of Fig. 3 (last paragraph of Sec. 4.2).*

Otherwise the implied goals are commendable. It would certainly be interesting to read about a new method of tree–ring climate reconstruction that does not involve transforming raw measurements into dimensionless indices, retaining the original units (e.g., Helama, 2015), and it would certainly be informative to learn what the "true color" of an MXD chronology is. However, I am not sure the method put forth in this study does any of this;

in fact I am not exactly sure what this method does other than rescales averaged, variance adjusted MXD measurements.

*The method used in this study is not supposed to be the "perfect" reconstruction method that retains the true color of climate. We analyse the effect of missing measurements and data pauses through computing the error variances of the annual climate indices over the whole data period. The variances are converted into degrees of freedom that are related to the number of measurements in each year and further to the quality of the climate indices.*

*In revised Sec. 4.3 it will be explained that the variance results are more generally valid:*

*"As far no new trees are added, the distribution of the measurements is not changed. Hence $S^2$ in Eq. (6) will be decreased but $Var_{rel}$ in Eq. (7) is unchanged. The latter term describes the error source studied here and depends only on the distribution of the measurements. Accordingly, it is more generally valid and is independent of developments in the reconstruction methods and their age dependence functions."*

How the results presented here lead to the conclusion there is a spectral bias, vis–à– vis FK2013, in the 1992 Torneträsk MXD chronology (BF92), thereby corroborating FK2013, is beyond my ability to detect. There are no spectral analyses performed, no modeling of persistence, and above all no hypothesis testing with statistical rigor of any kind. There are only graphical comparisons (wiggle matches) between chronologies and reconstructions.

*Admittedly we did not repeat how the bias was detected. Instead we gave only a reference in the introduction.*

*Different Torneträsk analyses (MK 2013) show a similar long–term variation (as far no new trees are added to the sample). Westward and eastward from the Torneträsk study area there are no corresponding long–term oscillations in the nearby reconstructions. Such local deviations at Torneträsk (of the order of magnitude of the greenhouse warming) are climatologically impossible and therefore the Torneträsk analyses must contain a bias. It follows that the Torneträsk case is suitable to illustrate results of our error analysis. Further, our reconstruction method should show the error variances and should in no way try to damp out the impact of the error sources.*

*- The detection of the bias will be explained in Sec. 4.1 of the revised manuscript as follows.:*

*"In Rinne et al. 2014 it was observed that both in the nearby oceanic (August SST, Norwegian Sea, Miettinen et al. 2012) and continental (Esper et al. 2012) temperature estimates of the long term oscillations clearly and similarly differ from those derived from the Torneträsk data. Such local anomalies are climatologically impossible and therefore the mutually similar long-term oscillations in the Torneträsk reconstructions contain a bias. Accordingly, such reconstructions are suitable for our error studies.*

*The differences observed are extreme being of the order of magnitude of the greenhouse warming. The long-term bias in Torneträsk reconstructions is thus detected climatologically. In our computations we estimate that climatological bias as the difference between the Torneträsk reconstruction and the corresponding Esper et al. (2012) reconstruction, the latter having a high number of trees"*

In this review I will argue the paper has no merit because i) the study is biased and provides no proof of significance, ii) the study is based on outdated data and does not contribute to advancing knowledge, iii) the proposed reconstruction method is does not produce a significantly different chronology, iv) the method does not account for inherent growth trends in MXD data, and v) the validation exercises are inconclusive.

I will conclude with a summary describing what I feel is the salvageable merit of this study, and a final comment on the gap-filling procedure described within.

**Comments re: RN()**

i) The scientific method and researcher bias

In science the null hypothesis defines a condition or relationship that an experimenter wishes to study and test. In a quest to find a difference between two conditions, A and B, the null hypothesis would be; there is NO difference. If the conditions in question have

quantities that can be measured then statistical tests are used to decide whether to accept or reject the null hypothesis. The decision to reject the null hypothesis, ergo there IS a difference, is based on the probability (significance) that the observed difference cannot be explained by chance alone. In RN() the implied null hypothesis is: the BF92 MXD chronology IS different than that produced here; however, there is no evidence that the observed difference is significant.

*The reconstruction produced in our study may contain low frequency bias, too. As mentioned in the text, the aim of this study in not to create a better reconstruction but only to estimate errors introduced by the time-varying tree-age distribution. Error sources should not be damped out so that the data will be close to the original one studied in BF92.*

There are libraries full of literature describing methods of signal processing and analysis of time series (e.g., Blackman and Tukey 1958, Percival and Walden 1993, Park 1992, Thomson 1982) that one can use to describe and compare the spectral properties of two time series (e.g., FK2013). Consider, in BF92 there is a plot of the power spectra of the MXD reconstruction produced using RCS (figure 9: BF92) (fig.1). Why couldn't the current authors have done the same? Or even push the concept further and computed the cross-spectral coherence between the two chronologies (http://www.spectraworks.com/web/welcome.html).

[Figure]

Figure 1 (BF92: fig.9b: reprinted without permission). The power spectra of the MXD reconstruction "based on the 300-lag autocovariance function and individual estimates have been smoothed with the Hamming window and have 12 degrees of freedom. The null continuum and 95% significance levels (for pre-defined peaks) are also shown".

Considering the ramifications of this study, and the overt claims of "erroneous bias", one would expect to find some statistical evidence for rejecting the unbiased, null hypothesis i.e., BF92 = RN(), but we don't. This is disconcerting to me for it means there really is no hypothesis testing and it is only the investigator's word that we must accept. Given the title and the exhausting use of bias and erroneous ("bias" is used 28 times; erroneous 10), it does not take a great deal of imagination to guess what the authors will conclude. The lack of a null hypothesis, and any attempt in applying statistical rigor to results, negates the significance of the conclusions.

*As explained above, we do not want to prove that our reconstruction is different from BF92 but rather have a similar reconstruction that allows us to study the error sources.*

ii) Why Briffa et al., 1992?

Since 1988 when the first Torneträsk MXD chronology was developed (Schweingruber et al., 1988) there has been tremendous effort and study invested in producing millennial length chronologies and reconstructions from the Scots pine trees in Fennoscandia, particularly those surrounding Lake Torneträsk, Sweden (Esper et al., 2014 and references therein). The most relevant of these is the most recent Melvin et al., 2013 (hereafter: MK2013).

With the exception of the last ~250 years when the Schweingruber et al., 1988 measurements are updated by the addition of predominantly faster growing, young living trees (Grudd 2002, 2008), the MXD data used in MK2013 are the same as those used in BF92, and this study. In other words, ignoring the post ~1650 CE period, MK2013 is essentially a re-analysis of BF92. That being the case then the real objective experiment would be to compare the chronology and reconstruction produced by RN() to that produced by MK2013. Why this was not done is again evidence to me that there is an a priori bias in RN(). So let us do it; let's compare MK2013 with RN() and decide which has more spectral bias.

*Exactly, the reconstructions have not been improved in the pre-1650. Thus, the error estimation of this study for the pre-1650 period is very relevant even in the most recent reconstruction. That newer studies improved the age distribution in the post-1650 period, where it was possible, is a clear sign that the scientific community is aware of the importance of the age-distribution error source. This study not only highlights and makes people aware of this error source but also quantifies its temporal variations.*

*Much of the progress from BF1992 to recent reconstructions was achieved by adding faster growing, young living trees is to a large part showing the improvements that could be achieved by improving the age distribution.*

Consider figure 3 in MK2013, reproduced here as figure 2. In panel

b we see the two chronologies produced from the high MXD (red) and low MXD (blue) value trees, along with their average. In addition the authors have kindly provided us with information on where in the two chronologies the sample size falls below 4 (thick and thin line widths); the source of those egregious "biases" the present study attempts to correct. For all practical purposes the black curve in panel c (One RCS Chronology) is effectively the BF92 chronology, and the red curve is the new, improved MK2013 Torneträsk chronology.

[Figure]

Figure 2. MK2013 main text figure 3 (reprinted without permission). "(a). The  black curve is based on all samples and the curves in red and blue were built from  samples with the highest and lowest values of MXD respectively, where sorting was  based on comparison of mean signal@free MXD against that of a single RCS curve  over their common period. b) shows mean chronologies created using two RCS  curves; for high@MXD samples (red), low@MXD samples (blue), and the average of all  samples (black). c) shows the chronologies created using a single RCS curve (black)  and two RCS curves (red). Chronologies were low@pass filtered using a 100@year  cubic spline. The thicker parts of the lines show sections of chronologies based on 4

or more samples and grey shading shows the sample counts over time."
Melvin et al., 2013.

By simply comparing the red and black curves in Fig.2b one sees
the new method proposed by MK2013 pays attention to temporal
changes in sample depth particularly during the Medieval period
where only the original Schweingruber et al., 1988 data are
contributing to chronology. By visually comparing the black and red
curves in Fig.2c one can imagine there is slightly less low frequency
variation in red curve than the black.

iii) Where's the bias

"The reconstructions in Fig. 3a are astonishingly close to each other,
in spite of the very different computational methods", RN(). So, is
there a problem to fix? Let's consider the results of a comparison
between BF92 chronology and the chronology produced by the
method proposed here in RN() figure 3 reproduced below also as
fig.3.

*- Comparison with Briffa et al. (1992) will be described in Sec. 4.1.as follows*

> *"In Fig. 3a the corresponding anomalies of the Torneträsk reconstruction
> in Briffa et al. (1992) are compared with our results. The climatologically
> biased quasi cycle of ca. 350 years is well seen. Especially the recent
> years are similarly biased showing an underestimation of the temperature.
> Excluding this recent underestimation, the variants of reconstructions in
> Melvin et al. (2013, their Fig. 4) show a similar long-term
> structure. Importantly, the bias is not in the tree ring measurements but in
> the reconstructions. Hence the bias has essentially remained as such from
> Briffa et al. (1992) until the present time, in spite of the developments in
> reconstruction methods. Naturally there can be differences in details
> because of different computational approaches. "*

[Figure]

5 37

10 8

69

42

**Figure 3. RN() Figure 3 with caption.**

"Figure 3. The theoretical explanation of biased long-term oscillations between 500 and 1975 AD. Panel (**a**): Smoothed (85-year spline) temperature reconstructions derived from Torneträsk MXD data 441-1980. Shown are the estimate from Briffa et al. 1992 and the present ones both with $b_{max}$=270 and $b_{max}$=370. Panel (**b**): The smoothed (85-year spline) difference between the present temperature estimate ($b_{max}$=270) and that in Esper et al. 2012. Panel (**c**): Comparison of the sample count (number of observations) and degrees of freedom. Indicated are the seven and two cases of DOF<5 and DOF>14, respectively. Panel (**d**): components of relative variances. Some years mentioned in the text and shown in Fig. 1 are indicated in panels (**b**), (**c**) and (**d**)."

RN(). In figure 3a I have marked the maximum and minimum levels of 14 major peaks and troughs in the two chronologies. For the purpose of example I used the b<=270 chronology to represent the "best" RN() chronology. It does not take a well-trained eye to see that the quasi-centennial scale variability in the b<=270 chronology is actually greater than in the BF92. For instance, between peaks 1 & 2 (fig.3a) the amplitude of the change in the b<=270 chronology is larger than in BF92, and between peaks 9 & 10, is larger still. Therefore, if there is a spectral bias in either of these two chronologies one could easily argue that it is greater in the b<=270 series.

"... it is greater in the b<=270 series." *That is true. In one earlier version of our manuscript it was mentioned that the two curves in our fig. 3a differ in amplitude but that the basic long-term structures are similar. It is natural that they can differ because of different computational methods. The comment was later removed. We add it now back following your analysis.*

*The main thing is that the different analyses show a similar long-term variation as far no new trees are added to the sample (as you*

*say:* In other words, ignoring the post ~1650 CE period, MK2013 is essentially a re–analysis of BF92").

*Fig. 4 from MK2013 has been attached in the following.*

iv) The fly in the ointment

MXD measurements have inherent growth trends and are commonly detrended by computing residuals, as opposed to ratios for tree ring–width (TRW) measurements, between the measured density and some mean biological growth function, viz. eq. 1.

$I_t = R_t - G_t$ eq.1

where R(t) is the MXD measurement for a given year, G(t) the value of some function chosen to best model the overall trend in R, and I(t) the resulting index for year t for t=1,age. As described in BF92, G(t) is commonly a smoothed version of the mean age–aligned biological growth function. In general density measurements range from .5 to .7 density units, and in their raw form MXD measurements are much more homoscedastic then TRW measurements (Figure 4). As Figure 4 clearly shows, there is still an age– related trend in MXD measurements that is not climate related and must be accounted for. [Incidentally, please check all references. I am sure the placement of Cook and Peters 1997 is not correct; as it pertains to solely TRW data and the calculation of tree–ring indices as ratios, making it tangentially relevant to our discussion here but not where it is in the main text. I believe the more appropriate reference for the main text is Cook et al.,1995)].

*We will add a reference to* Cook et al.,1995.

The fact that MXD data have demonstrably less trend than TRW measurements is the reason RN() could simply average MXD measurements and produce a plausible chronology. I suspect that had RN() accounted for the non–climatic trend in the MXD measurements then their resulting chronology would have less "bias"

and quite possibly look more like BF92.

[Figure]

Figure 4. Growth trends in TRW and MXD measurements. Panel a, the age aligned trends: Regional Curves, panel b the calendar aligned trends (reprinted without permission from https://www.researchgate.net/figure/277853533_fig2_Figure-2-TRW-and-MXD-age-trends-a-Arithmetic-means-of- the-age-aligned-TRW-and-MXD)

v) Let's be fair

A perceived erroneous bias in the first multi-centennial MXD reconstruction based on RCS done 25 years ago does not demonstrate the existence of "universal error" in all RCS reconstructions. Such a statement is utterly fanciful and defamatory without proof.

*We understand that the title may sound to bold and changed it consequently. However, time varying age distribution commonly lead to errors and are a known and existing problem. Especially far back in time, age-distribution changes cannot be avoided by a good*

*sampling strategy because there are simply no samples available. With our method we identify periods, when this is a serious problem.*

The claimed illustration of bias in the Torneträsk reconstruction through comparison with the NSCAN MXD–RCS temperature reconstruction (Esper et al., 2012) in figure 3b also has problems that need to be at least admitted too. The assumption made is that the reconstruction of Esper et al. (2012) is in some sense more representative because it is based on a much larger sample size that is likely to be less affected by age–class gaps and paucities. The problem with this comparison is that the MXD data used in the Esper et al. (2012) reconstruction contains a significant amount of material from Finland, including sub–fossil lake material which the 1988 and 1992 Torneträsk chronologies do not. That is clearly indicated in Fig. 1 and Table S1 of Esper et al. (2012) and also in the main text of Esper et al. (2014).

In figure 3b, the use of completely different data, without a defendable argument for why this is okay, is not good science and undoubtedly contributes to some of the lack of agreement. In figure 3b a slightly better choice would be the N–Eur reconstruction Esper et al. (2014) as this reconstruction recognizes the affect of using trees of varying age classes.

*The presented general theory is illustrated with the aid of the Torneträsk data. The error source introduced in our article describes the dependence on the distribution of the measurements over the years and age classes. There clearly are other error sources but they do not affect our case. The bias observed in the Torneträsk case is well explained by the error source studied, i.e. by data gaps and paucities in the data.*

*- The explanation of the bias with the aid of the error source studied will be given in the revised Sec. 4.2.:*

*"To illustrate the combined impact of different error terms, the degrees of freedom are computed with the aid of the approximate formula in Eq. (7) and*

*are shown in Fig. 3c. In the beginning of the data, they grow slower than the number of measurements (Fig. 3c), show rapid changes thereafter and are often very low. Main minima and maxima of the DOF are indicated by full and open circles, respectively. Fig. 3b shows the bias estimate of the reconstruction with $b_{max}=270$. It is seen, that strong minima (maxima) of the DOF in Fig. 3c well predict following biased (unbiased) values in Fig. 3b and fit with the anomalies in reconstructions shown in Fig. 3a.“*

**Summary**

Throughout this manuscript there were enough confusing comments and descriptions regarding previous work and dendrochronological methods that lead me go back and look at the author's affiliations. That's when it struck me as to what Rinne et al. () are trying to do  They are treating the Torneträsk MXD data as if the data were a group of meteorological records with periods of missing data, filling those gaps, and using the Tornedalen historical record to rescale the result. This sounds a lot to me like homogenizing data.

However, what strikes me as the most egregious non-scientific element in this work is found in following extract.

"Klingberj [sic] and Moberg (2003) composited a series of instrumental observations made in the Tornedalen region, some 300 km from Torneträsk. Their construction begins with instrumental data from Övertorneå (1802–1838). The observation hours were not stated explicitly for 1826–38 and the authors had to assume them. Here we correct that assumption by increasing their JJA temperature reconstructions by 1°C. The procedure can be interpreted as if an unknown component in the instrumental observations had been deduced from the tree ring estimates. Nevertheless, temperatures before ca. 1850 may still contain biases (Melvin et al., 2013; Grudd, 2008)."

If I understand this correctly, RN() are correcting an assumption using tree- ring estimates while at the same time claiming the treering estimates are biased? This boggles my mind. Not only does this confirm my opinion on how well long–historical records can be used for the assessment of spectral bias (e.g., FK2013, Osborn and Briffa 2003), but nails the coffin shut on my opinion of the present work. It completely explains how RN() could again produce a plausible reconstruction using their method  There are so many vagaries is this story that I feel the need for a new word for bias.

*On the basis of your comment, the background of the correction made is now discussed in more details. Both the original and corrected versions are now shown in Fig. 4.*

*Discussion will be added to Sec. 3 (Torneträsk case study, changes in blue) and Fig. 4 has been changed adding the original data (dotted line):*

*Klingberj and Moberg (2003) composited a series of instrumental observations made in the Tornedalen region, some 300 km from Torneträsk. Their construction begins with instrumental data from Övertorneå (1802-1838). The observation hours were not stated explicitly for 1826-38 and the authors had to assume them. The JJA temperatures are known to be sensitive to the observation hours used. If these are not precisely known, true climatic variations may remain but the average level of the temperature may become wrong. In such cases any kind of support is welcome. During 1826-38 the bias estimate in our reconstructions is rather small (≈ -0.25°C, Fig. 3a) and the DOF are rather high (≈11, Fig. 3c).  Our reconstruction can therefore be applicable. If the JJA temperatures in observations are systematically increased by 1°C every year during 1826-38, the smoothed result happens to fit rather well with our reconstruction in Fig. 4. Accordingly, the corrected temperatures are supported by the reconstruction and the anomalous coldness in observations before 1840 in Fig. 4 seems suspicious due to the unknown observation hours. The comparison with instrumental data here - while interesting  - is not essential, moreover as in any case temperatures before ca. 1850 may contain biases (Melvin et al., 2013; Grudd, 2008).*

[Figure]

Figure 4. Comparison of Torneträsk temperature reconstruction and Tornedalen temperature observations 1802-1977. Reconstruction mean has been adjusted to fit the corresponding observational mean during 1850-1950 and both curves have been smoothed.

I strongly disagree this work brings any new insights into the causes of spectral bias in climate reconstructions from tree rings. In fact I would argue it demeans the FK2013 argument. The appearance of long–memory in long tree–ring reconstructions is just as likely to reflect the fact that climate has varied on a variety of time scales over the past millennia (NRC 1991, Koutsoyiannis 2002) and that our extant historical records of climate are too short to model this condition. Accepting the fact that we are not likely to find any new historical records, we must do the best we can to explore the proxy record. This study does not do that at all.

The writing style and structure of the manuscript could certainly be

improved. I am sure I have misunderstood a meaning or two here and there simply because I could not understand clearly what was written.

*We wish that the clarifications and explanations to be added into the manuscript make the text clearer.*

The one potential merit is the interpolation/extrapolation method described in section 2.1 and illustrated in Figure 2:RN(). The question of how a non- stationary distribution of age-classes affects a chronology is one Dendrochonologists continually revisit (google any of the more recent RCS publications). I highly recommend starting with Esper et al., 2003 " Test of the RCS method for preserving low-frequency variability in long tree-ring chronologies". As the title implies, Esper et al., 2003 is obviously germane

to the topic in hand and should have been referenced.

The questions I ask are, what do the simulated values look like, and is this step really necessary? I wonder how biologically relevant are the simulated values? This is certainly an area in dendrochronology that could benefit from further research, but more in the vein of Esper et al., 2003 rather than this.

*Our reconstruction method has been designed to give estimates of error variances that can be used to analyze error sources. It is enough if the reconstruction method otherwise works sufficiently satisfactorily.*

*- Explanatory text on the performance of our method will be included in Sec. 4.3.:*

> *"The reconstruction method has been designed for estimation of the error variances. A general form of variances is found in Eq. (5). This is applicable as well for tree ring widths as maximum latewood densities. Only three parameters are needed in explicit applications ($r_1$, $S$, $r_n$). Error sources are not corrected but retained in order to study them. For instance, the long-term erroneous*

*oscillations in Briffa 1992 are well reproduced in Fig. 3a during recent years, too. As a summary, the reconstruction method performs as it should.*

*The reconstruction method gives two values for every year, an average and its error estimate. If the parameters ($r_1$, S, $r_n$) were known, yearly confidence limits of the reconstruction could be given. An important part here is the variances due to missing young and old age classes.*"

Finally, something I just noticed, the age-class limit found optimal in this study ($b_{max}$=270) is remarkably close to the ideal age range used in Esper at al., 2014.

*That has not been mentioned in our manuscript as Esper et al. did not describe the way by which they selected 306. Of course it is better to give a reference. We add it.*

"For the final summer temperature reconstruction we only used tree rings of a certain biological age ranging from <=31 to >=306 years, i.e. removed the rings 30 years (here termed as the 'young rings') and >306 (here termed as the 'old rings') from the combined, and adjusted S88 + E12 dataset." (Esper et al., 2014)

In the quote above S88 is the Schweingruber et al., 1988 dataset, E12 is the 587 MXD collection used in Esper et al., 2012. The similarity between the two upper limits of age class restriction further convinces me the present work provides little insight in the way contemporary Dendroclimatology applies Regional Curve Standardization.

-pjk Stockholm

Attachment:

Fig. 4 in Melvin et al. 2013 has been referred to in the manuscript. It is attached here (https://crudata.uea.ac.uk/cru/papers/melvin2012holocene/).

There are several reconstructions from several years (e.g. Briffa MXD 1992 red+green in the uppermost panel, Grudd TRW 2002 blue in the next one, Briffa TRW 2011 blue; recent ones in the three lower panels). All of them show similar long-term oscillations that are seen in

Briffa 1992 MXD. It is seen that recent advanced reconstruction methods have not changed the situation.

Exceptions are only seen during recent years where the "divergence" problem is best seen in Briffa 1992 MXD (green line). There are several attempts to correct the problem. No similar variations in the "divergence" case (systematic underestimation of the temperature) after 1600 AD are visible. The cold bias is similarly in different reconstructions of the order of magnitude of the greenhouse warming.

[Figure]

**Changes to the manuscript.**

*Contents*

**2.2 Explicit computation formulae**

*(page 4, lines 9-23, the older version of the following new paragraph began with "Assume …")*

In order to get the average taken over all age classes, the contributions of the missing young and old age classes are needed. These can be estimated as follows. First estimates of measurements are interpolated for every age class where that is possible. Then years with values in age classes of $b$=20 and $b$=270 are selected. This makes it possible to compute yearly averages over all age classes of 20 thru 270. Next yearly averages over age classes of 21 thru 270 are computed. Generally this new average is smaller but the variation between the years is strong. By computing the mean and r.m.s.e. of the yearly differences between the averages with and without b=20 we get an estimate of the eliminated age class $b$=20. In the next step the yearly averages over age classes of 22 thru 270 are computed and used to estimate the impact of missing age classes of $b$=21 and $b$=22 to the average of all measurements between $b$=20 and $b$=270. The computations are continued by dropping out more age classes. Similar approach is applied to estimate the impact of missing old age classes.

The estimation turned out to be more complicated if only very young age classes ($b_1<\approx20$) or most of the older age classes ($b_n<30$) were missing. To keep the formulae simple, the linear approximation is extended to those cases, too.

The result is that the contributions of the missing young and old age classes to the average taken over all age classes can be estimated linearly by $M_{young}\approx r_1(b_1-1)$ and $M_{old}\approx rn(bn-bmax)$, respectively, where $r_1\approx0.000128$ and $rn\approx0.000170$. Note that $M_{old}$ results in a negative correction. The accuracy of such corrections turned out to be low their variances being roughly $[r_1(b_1-1)]^2$ and $[rn(bn-bmax)]^2$. The application of the corrections are illustrated for an individual year in Fig. 2 for $bmax$=300.

The focus here is in the long-term error. Important are the estimates of the error variances due to the missing age classes. Precise values of $r_1$ and $r_n$ are not needed. The main information included in the results is the dependence of the variance terms on the lengths of the data gaps, $b_1$-1 and $bmax$-$bn$. Our reconstruction method is directed to uncover that dependence explicitly in a simple way. Otherwise it is enough that the reconstruction method performs sufficiently satisfactorily. It needs not be an optimal one.

No assumption on the age function is made. The contributions of the young and old age classes can widely vary between years as indicated by their error variances. However, average corrections are needed in order to decrease the impact of systematic errors. By adding the corrections, the yearly average of the interpolated, extrapolated and measured values becomes approximately **...**

**3 Torneträsk case study**

*(page 7, lines 1-6, the changes are given in blue)*

Klingberj and Moberg (2003) composited a series of instrumental observations made in the Tornedalen region, some 300 km from Torneträsk. Their construction begins with instrumental data from Övertorneå (1802-1838). The observation hours were not stated explicitly for 1826-38 and the authors had to assume them. The JJA temperatures are known to be sensitive to the observation hours used. If these are not precisely known, true climatic variations may remain but the average level of the temperature may become wrong. In such cases any kind of support is welcome. During 1826-38 the bias estimate in our reconstructions is rather small (≈ -0.25°C, Fig. 3a) and the DOF are rather high (≈11, Fig. 3c). Our reconstruction can therefore be applicable. If the JJA temperatures in observations are systematically increased by 1°C every year during 1826-38, the smoothed result happens to fit rather well with our reconstruction in Fig. 4. Accordingly, the corrected temperatures are supported by the reconstruction and the anomalous coldness in observations before 1840 in Fig. 4 seems suspicious due to the unknown observation hours. The comparison with instrumental data here - while interesting - is not essential, moreover as in any case temperatures before ca. 1850 may contain biases (Melvin et al., 2013; Grudd, 2008).

*(corrected and original observations are now shown in Fig.4)*

[Figure]

Figure 4. Comparison of Torneträsk temperature reconstruction and Tornedalen temperature observations 1802-1977. Reconstruction mean has been adjusted to fit the corresponding observational mean during 1850-1950 and both curves have been smoothed.
* * *
*(revised version of the discussion, page 7, line 17 – page 9, line 7)*

**4.1. Bias of the Torneträsk reconstructions**

In Rinne et al. 2014 it was observed that both in the nearby oceanic (August SST, Norwegian Sea, Miettinen et al. 2012) and continental (Esper et al. 2012) temperature

estimates of the long term oscillations clearly and similarly differ from those derived from the Torneträsk data. Such local anomalies are climatologically impossible and therefore the mutually similar long-term oscillations in Torneträsk reconstructions contain a bias. Accordingly, those reconstructions are suitable for our error studies.

The differences observed are extreme being of the order of magnitude of the greenhouse warming. The long-term bias in Torneträsk reconstructions is thus detected climatologically. In our computations we estimate that climatological bias as the difference between the Torneträsk reconstruction and the corresponding Esper et al. (2012) reconstruction, the latter having a high number of trees. In Fig. 3a the corresponding anomalies of the Torneträsk reconstruction in Briffa et al. (1992) are compared with our results. The climatologically biased quasi cycle of ca. 350 years is well seen. Especially the recent years are similarly biased showing an underestimation of the temperature. Excluding this recent underestimation, the variants of reconstructions in Melvin et al. (2013, their Fig. 4) show a similar long-term structure. Importantly, the bias is not in the tree ring measurements but in the reconstructions. Hence the bias has essentially remained as such from Briffa et al. (1992) until the present time, in spite of the developments in reconstruction methods. Naturally there can be differences in details because of different computational approaches.

**4.2. Explanation of the bias observed**

The error formulae derived are of a general form (Eqs. 4 and 5). The error variances depend on the relative error terms and the dimensional portions $(b_{max} \, r_1)^2$, $(b_{max} \, r_n)^2$ and $S^2$. Because the relative error terms depend only on the distribution of the measurements over the age classes and years, they can be applied to any data without knowing the measurements or reconstruction method used.

In Fig. 3 the relative variances are applied for the Torneträsk data with 65 trees. Paucities in the tree population or periods of missing trees (Fig. 1) cause successively data gaps in young, intermediate and old age classes leading to variance variations in Fig. 3d. The increased variance decrease the degrees of freedom and so makes the bias more probable, which is then seen as erroneous long-term oscillations in the reconstructions.

To illustrate the combined impact of different error terms, the degrees of freedom are computed with the aid of the approximate formula in Eq. (7) and are shown in Fig. 3c. In the beginning of the data, they grow slower than the number of measurements (Fig. 3c), show rapid changes thereafter and are often very low. Main minima and maxima of the DOF are indicated by full and open circles, respectively. Fig. 3b shows the bias estimate of the reconstruction with $b_{max}=270$. It is seen, that strong minima

(maxima) of the DOF in Fig. 3c well predict following biased (unbiased) values in Fig. 3b and fit with the anomalies in reconstructions shown in Fig. 3a.

Low values of the DOF (<5) are related to extrema of the bias. High values of the DOF (>14) indicate vanishing bias. The latter ones are not numerous and therefore bias-free cases are seen only temporarily in Fig. 3b.

One year with low DOF is separately indicated in Fig. 3. A change in the tree population is seen in 1356 AD (Fig. 1) where the oldest age class (<270 years) drops suddenly down to b=116 years. As a consequence, variances of the older and intermediate age classes are peaked in Fig. 3d, further lowering strongly the DOF in Fig. 3c. This is reflected in the bias (Fig. 3b) and in the reconstruction (Fig. 3a, smoothed values in Figs. 3a and 3b naturally lag the sudden changes in Figs. 3c and 3d). Correspondingly the optimal distribution of the measurements around 869 and 1500 AD (Fig. 1) are reflected in low error variances (Fig. 3d), high number of the DOF (Fig. 3c) and unbiased temperature estimates (Fig. 3b).

In Fig. 1, there is a repetitive similarity between the two longer periods with nearly continuous samples of trees (AD 1250-1500 and 1600-1800) and two periods without new trees (AD 1500-1600 and 1800-1980). The corresponding terms of the relative variance (Eq. 5) will be discussed in the following.
The large first peaks of variance around 1356 AD (Fig. 3d) result from missing intermediate and old age classes. These damp slowly out and new peaks are seen at 1600 AD. Here they are due to the intermediate (interpolated) and young age classes. The same structure is repeated between 1600-1800 AD and 1800-1980 AD. In this way the paucities due to missing trees in the data cause alternating data gaps leading to varying behavior in temperature reconstructions. Especially biases around 1600 and 1950 can be explained by the error terms connected to the missing younger age classes (due to the missing trees). The latter case is known as "divergence", a systematic underestimation of the temperatures (Briffa et al. 1998, D'Arrigo et al. 2008). As there is no formal difference between the cases of 1600 and 1950, it is natural to refer to "divergence" in both cases.

**4.3 The performance of the reconstruction method applied**

The reconstruction method has been designed for estimation of the error variances. A general form of variances is found in Eq. (5). This is applicable as well for tree ring widths as maximum latewood densities. Only three parameters are needed in explicit applications ($r_1$, $S$, $r_n$). Error sources are not corrected but retained in order to study them. For instance, the long-term erroneous oscillations in Briffa 1992 are well reproduced in Fig. 3a during recent years, too. As a summary, the reconstruction method

performs as it should.

The reconstruction method gives two values for every year, an average and its error estimate. If the parameters ($r_1$, $S$, $r_n$) were known, yearly confidence limits of the reconstruction could be given. An important part here is the variances due to missing young and old age classes.

To illustrate, the Torneträsk data is applied. Figs. 3a and 4 show that the reconstruction method performs sufficiently satisfactorily and so the error estimates can be seen to be reasonable. On the other hand, the relative errors due to data gaps and paucities are seldom low (e.g. years 869 and 1500 in Fig. 3d). Therefore the reconstruction can be expected to be inaccurate during most years.

If the errors due to the distribution of the measurements are not taken into account, in Eq. (6) $Var_{rel}$=1. Conventionally the sample accuracy is then characterized by presenting the yearly number of measurements. This practise is followed here. Instead of trying to estimate confidence limits, the sample accuracy is characterized by giving the degrees of freedom (Fig. 3c). The practical application requires that the general error terms in Eq. (5) are approximated and compressed into Eqs. (6) and (7).

In the recent literature new methods to estimate the age dependence function more precisely are introduced (e.g. Briffa et al. 2013; Melvin et al. 2013; Matskovsky and Helama 2014). They improve the reconstruction and the errors in the long-term oscillations may be decreased. Potential error sources in the sampling technique have been detected (e.g. Bowman et al. 2013). As far no new trees are added, the distribution of the measurements is not changed. Hence $S^2$ in Eq. (6) will be decreased but $Var_{rel}$ in Eq. (7) is unchanged. The latter term describes the error source studied here and depends only on the distribution of the measurements. Accordingly, it is more generally valid and is independent of developments in the reconstruction methods and their age dependence functions.

In our reconstruction method the missing younger and older age classes are taken into account. It is natural that the results resemble to those that use age dependence functions. However, here the impact of the missing younger and older age classes is estimated directly from the measurements and only for the average impact of the missing age classes. The resulting estimates of error variances are high in the cases of data gaps and paucities. Therefore there is here no need for a more advanced reconstruction method because necessarily the error estimates will be high and the estimated impact of missing age classes less accurate.

There is one possibility in some specific cases to regulate the accuracy. Our method implies that an upper limit of the age classes is selected. The higher is the limit, the more measurements are included into the analysis. Simultaneously there will be more longer data gaps. It is to find a compromise between the opposite effects. An illustration is given in Fig. 1, where the cases of $b_{max}$=270 and $b_{max}$=370 are compared. In the Tornedalen case study, it was concluded to make use of $b_{max}$=370 in order to decrease the bias during recent years. That choice was motivated as the hatched line in Fig. 3a is closer to zero. Otherwise it is seen that the changes are weak and the selection of $b_{max}$ is not decisive. If it wished to be made mechanically, a possibility would be to minimize $Var_{ave}$ with respect to $b_{max}$ in Eq. (7). In Esper et al. (2014), the upper limit was taken to be $b_{max}$=306.

---

## Author Comment (AC5) · 5 Jun 2016

Y. Markonis

In lines 32-33 the authors claim that: "In fact, Franke et al. (2013) succeeded in showing that many present day state-of-the-art reconstructions still tend to be biased in the low frequency part of the spectra." However, more recent findings (Iliopoulou et al. 2016; Markonis and Koutsoyiannis 2016) suggest that this may not hold true. Following the approach of Bunde et al. [2013], in determining the low frequency bias by examining the long-term persistence behaviour of precipitation, it has been shown that the low frequency variability is evident in many different types of proxies and not only to the tree-rings. In addition, a simple explanation was provided for this behaviour based

on the changing dependence structure of precipitation as the temporal scale increases [Markonis and Koutsoyiannis 2016].

Thank you for your comment. We will add a reference to your paper as follows (lines 4 and 5 of from "Reasons" to "strongly"):

Franke et al. (2013) observed that the low-frequency biases can be present in studies applying maximum densities of the tree rings as well as those studying tree ring widths. The bias can further be present both in estimates of precipitation and temperature. Reasons are currently a topic of scientific discussion, e.g. a link between record length and persistence has recently been proposed (Markonis et al. 2016). Our study strongly suggests that the paucities in the data are one explanation for such a general presence of the error. The approximate method derived here to estimate the degrees of freedom can be used to trace the potential impact of such paucities.
* * *